# Parasitic capacitance modeling and measurements of conductive yarns for e-textile devices

Ziqi Qu [1,2], Zhechen Zhu[1,3], Yulong Liu [1,4], Mengxia Yu [1,5] & Terry Tao Ye [1] ✉

Conductive yarns have emerged as a viable alternative to metallic wires in e-Textile devices, such as antennas, inductors, interconnects, and more, which are integral components of smart clothing applications. But the parasitic capacitance induced by their micro-structure has not been fully understood. This capacitance greatly affects device performance in high-frequency applications. We propose a lump-sum and turn-to-turn model of an air-core helical inductor constructed from conductive yarns, and systematically analyze and quantify the parasitic elements of conductive yarns. Using three commercial conductive yarns as examples, we compare the frequency response of copper-based and yarn-based inductors with identical structures to extract the parasitic capacitance. Our measurements show that the unit-length parasitic capacitance of commercial conductive yarns ranges from 1 fF/cm to 3 fF/cm, depending on the yarn's microstructure. These measurements offer significant quantitative estimation of conductive yarn parasitic elements and provide valuable design and characterization guidelines for e-Textile devices.

E-textile technology had demonstrated its unique advantages in smart clothing and wearable device applications, where conductive yarns play an important role in the construction of many electronic components, such as inductors[1], capacitors[2,3], sensors[4–6], interconnects[7], and antennas[8–11], etc. Compared with traditional flexible electronics fabricated on PI or PET substrates, e-textile devices fabricated with conductive yarns not only exhibit superior flexibility to accommodate human daily movements (such as twists, inflects, or stretches), their esthetic appearance and comfortability also make e-textile devices the ideal solution for long-term medical monitoring and tracking in patient[12] and elderly care services[13].

Conductive yarns can be categorized into three types according to their constituent fibers: pure electrically conductive metallic fibers (PECM, such as stainless-steel)[14], intrinsic conductive polymer (ICP) fibers[15], and conductive polymer composite (CPC) fibers[16,17]. Through proper choice of materials, conductive yarns with different mechanical and electrical properties can be catered to accommodate multiform

applications, where different wearable devices have different requirements for conductive yarns[18,19]. Table 1 lists some typical e-textile devices along with the conductive yarns used for construction[7,20–25]. Conductive yarns are not a simple replacement of traditional metallic materials, i.e., the intrinsic electrical and mechanical properties of conductive yarns led to many challenges in the design and fabrication of these devices. For example, conductive yarns normally have much higher resistance as compared to metallic counterparts; inferior conductivity leads to the increase of energy dissipation and lowered quality factor (Q-factor). Moreover, yarn-based structures cannot maintain fine geometries with higher resolution; e-textile devices fabricated with conductive yarns have to adopt coarser geometries and simpler construction methods.

However, many e-textile designs only regard conductive yarns as a substitute of metallic wires; these designs often adopt traditional ways of circuitry construction, only to use conductive yarns as an alternative for metallic materials. More recent research began to investigate the

[1]Department of Electrical and Electronic Engineering, Southern University of Science and Technology, Shenzhen 518055, China. [2]Department of Nanotechnology, University of Pennsylvania, Philadelphia, PA 19104, USA. [3]Department of Electrical Engineering, University of Pennsylvania, Philadelphia, PA 19104, USA. [4]Department of Applied Physics, The Hong Kong Polytechnic University, Hong Kong, China. [5]Department of Electrical and Computer Engineering, National University of Singapore, Singapore, Singapore. ✉e-mail: yet@sustech.edu.cn

**Table 1 | Conductive-yarns-based devices applications**

| Reference | Applications | Yarns used | Limitations |
|---|---|---|---|
| 7 | Interconnects | Copper and Metallic Yarn | Lower Conductivity Robustness to Deformation |
| 20,21 | Strain Sensor | Conductive Polymer/Carbon Coated Yarns | Lower Sensitivity and Dynamic Range |
| 22 | Wearable Antenna (for Wireless Rx/Tx and RFID) | Stainless Steel Yarns/Silver Coated Fibers | Antenna Geometry Resolution and Lowered Q-Factor |
| 23 | Inductive and Coupling Coils (for NFC and Energy Harvesting) | Copper Strand Twisted on PET Filaments/Silver Coated Yarns | Antenna Geometry Resolution High Resistivity and Lower Q Factor Parasitic Elements |
| 24 | Piezoelectric Sensor | Piezoelectric poly-l-lactic acid (PLLA) fibers | Large Impedance, High Noise, Unsensitive with Specific Structures |
| 25 | Pressor Sensor | Silver-plated nylon fiber coated with polyester | low sensitivity, low robustness |

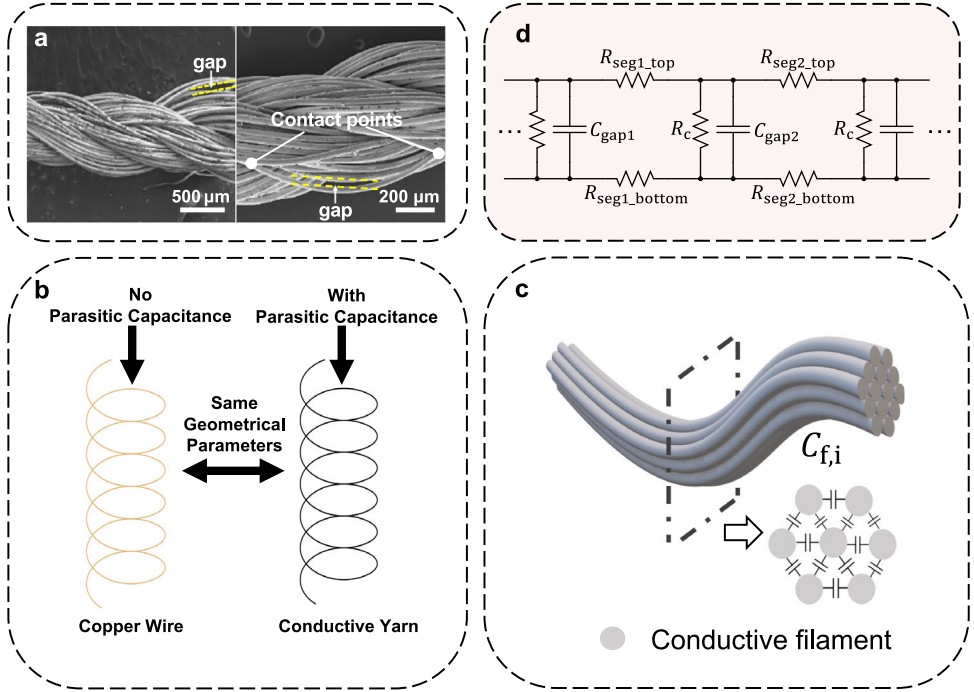

**Fig. 1 | Micro-structure of conductive yarn and existence of parasitic capacitance. a** Microstructure of the conductive yarns where gaps are formed between the twist filaments. **b** Estimation of the parasitic capacitance of conductive yarns through the comparison of reactive performance of two helical inductors with the same geometrical parameters (one constructed from copper (left), one from conductive yarns (right)). **c** An equivalent circuit model of parasitic capacitances formed by the gaps between adjacent filaments, and **d** Cross section of the conductive yarn: parasitic capacitances are uniformly distributed along the conductive yarns.

unique design challenges of e-textile devices[26–28]. However, these works mostly focus on the resistive properties of the conductive yarns, the reactive properties, which determine the devices' performance in high frequency and RF applications (typically at frequencies above 1 MHz), had not been systematically investigated.

We speculate that in addition to the intrinsic resistance, conductive yarns also possess intrinsic parasitic capacitance that affects e-textile devices' reactive performance. The parasitic capacitance is induced by the twisting structure of filaments in the construction of conductive yarns. Figure 1a shows the microstructure of a thread of archetypical conductive yarn under Scanning Electronic Microscope (SEM). From the image, we can see gaps are unavoidably formed between filaments when they are twisted to construct yarns. The filaments and the gaps in between constitute a charging-plate structure and hence create parasitic capacitances. As far as the authors are aware, no previous studies have attempted to create circuit models and estimate specifically the parasitic capacitance of conductive yarns, despite the significant impacts that the parasitic capacitance may have on e-Textile devices. Nevertheless, it is also difficult to precisely quantify the value of parasitic capacitance just by simply using multi-

meters or a network analyzer. In this paper, we propose a lump-sum model and a turn-to-turn model to estimate the parasitic capacitance of conductive yarns, together with a systematic method to extract and estimate these two forms of parasitic capacitances. The method needs to construct two helical air core inductors of the same geometry as shown in Fig. 1b, i.e., same diameter, same turn-to-turn separation and same number of turns. One helical inductor is winded using conductive yarns, while the other is winded using copper wires of the same gauge (diameter of the wires). While the lump-sum capacitance as well as the turn-to-turn capacitance are created by the helical structure intrinsically (we call them the structural capacitance in this paper and they are measured from the copper wire inductor), the parasitic capacitance from conductive yarns will contribute additional capacitance to the structural capacitances. By comparing the resonant frequency of these two helical inductors, the yarns' lump-sum parasitic capacitance can be measured. Moreover, through comparing the reactant curves at different number of turns of these two helical inductors, the turn-to-turn parasitic capacitance of conductive yarns can also be extracted. The lump-sum parasitic capacitance and turn-to-turn parasitic capacitance correlate to each other, and further verify

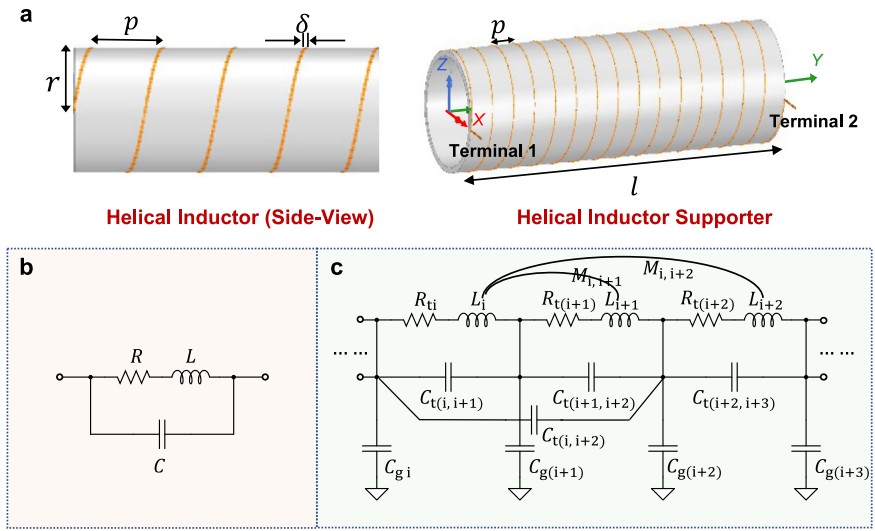

**Fig. 2 | Helical inductor designation and two main equivalent circuit. a** Structure of single-layer air-core helical inductor. **b** Lump-sum equivalent circuit of the helical air-core inductor. **c** Turn-to-turn equivalent circuit model of the helical inductors.

the accuracy of our parasitic capacitance models and the extraction method.

Using these two parasitic capacitance measurements, the unit-length parasitic capacitance can be derived. We have measured the parasitic capacitance of three different conductive yarns, namely, S310 (stainless steel yarn with diameter of 0.31 mm), S480 (stainless steel yarn with diameter of 0.48 mm) and AMBERSTRAND® 166 (AMBER-STRAND 166 Datasheet Available at: https://static1.squarespace.com/static/558431b9e4b0875de16c5494/t/5d9d011bb7bacf1e9e34d93c/1570570527783/Amberstrand+166.pdf.) (conductive polymer composite yarn with diameter of 0.25 mm), the unit-length capacitance (in term of fF/cm) are 2.71, 0.99 and 3.53. We selected these conductive yarns because they represent a wide range of categories within the conductive yarn market. It is also important to mention that, like many other parasitic parameter extraction techniques used for electronic devices. The extraction results are affected by many structural and ambient conditions and cannot be very accurate. Nevertheless, the techniques proposed in this paper can be easily applied to measure the parasitic capacitance of other yarns. Aided by our proposed lump-sum model and turn-to-turn model, the measurements can be used in the calculation and simulation of the performance of conductive yarn-based e-textile devices.

## Results
### Conductive yarn parasitic capacitance modeling
As seen from the SEM picture that shows the microstructure of a thread of conductive yarns, tiny gaps are formed between the twisting filaments. Because of the intrinsic resistance of the yarn materials, adjacent filaments may not have the same potential. To be more specific, as illustrated in Fig. 1a, when bundles of filaments are twisted, gaps and contacts are formed along the length of adjacent filaments. Filaments are resistive in nature and the length of adjacent filaments between contact points may not have the same length. The difference of resistance will lead to different distributions of voltage along the filaments, and will result in different potentials between adjacent filaments. Two filaments and the gaps in between can form a plate structure and create parasitic capacitances. As the gaps are randomly formed, the parasitic capacitance can be assumed to be distributed uniformly along the conductive yarns. A distributed equivalent circuit model of the parasitic capacitances created by the gaps is illustrated in Fig. 1c, where $R_{seg1\_top}$ and $R_{seg2\_top}$, etc. are the

resistance of the filament segments that form the top plate of parasitic capacitances between two contact points, and $R_{seg1\_bottom}$, $R_{seg2\_bottom}$, etc., are the resistace of the filament segments that form the bottom plate. $R_c$ is the contact resistance formed by the adjacent filaments touching each other at the ends of the gaps. $C_{gap1}$ and $C_{gap2}$ are the parasitic capacitances created by the gaps.

The capacitance formed by each tiny gap can be consolidated into parasitic capacitance between adjacent filaments. Figure 1d illustartes a cross section of the conductive yarn and the collective parasitic capacitances between the filaments. We denote the inter-filaments capacitance as $C_{f,i}$, which is formed between two filaments $f$ and $i$. For a unit length segment on the yarn, capacitance $C_{f,i}$ can be consolidated as a unit-length capacitance $C_p$, which is distributed in parallel along the length of the yarns.

### Lump-sum structural model of the helical inductor
Firstly, we introduce the lump-sum circuit model of the helical air-core inductor constructed from ideal metallic wires. Using ideal metallic wires implies that there are no parasitic capacitances induced by the microstructure of the wire. The helical structure can be defined by the number of turns $N$, diameter of the core $r$, the separation between the adjacent turns $p$, and the width of grooves $\delta$.

The lump-sum equivalent circuit model of the ideal helical inductor is depicted in Fig. 2b. The circuit forms a simple RLC resonator that consists of the total resistance $R$, total inductance $L$ and total capacitance $C$. Although the inductor is constructed from ideal metallic wires, the helical structure itself will generate capacitances, which include the turn-to-turn capacitance and turn-to-ground capacitance (will be discussed later). The lump-sum capacitance $C$ in the model consolidates these two structural capacitances. The resonant frequency $f_r$ of this RLC resonator can be calculated with Eq. (1) below.

$$f_r = \frac{1}{2\pi}\sqrt{\frac{1}{LC} - \left(\frac{R}{L}\right)^2} \tag{1}$$

### Turn-to-turn structural model of the helical inductor
Turn-to-turn model considers the structural parameters within each turn, which include the segmented resistance, the mutual capacitance

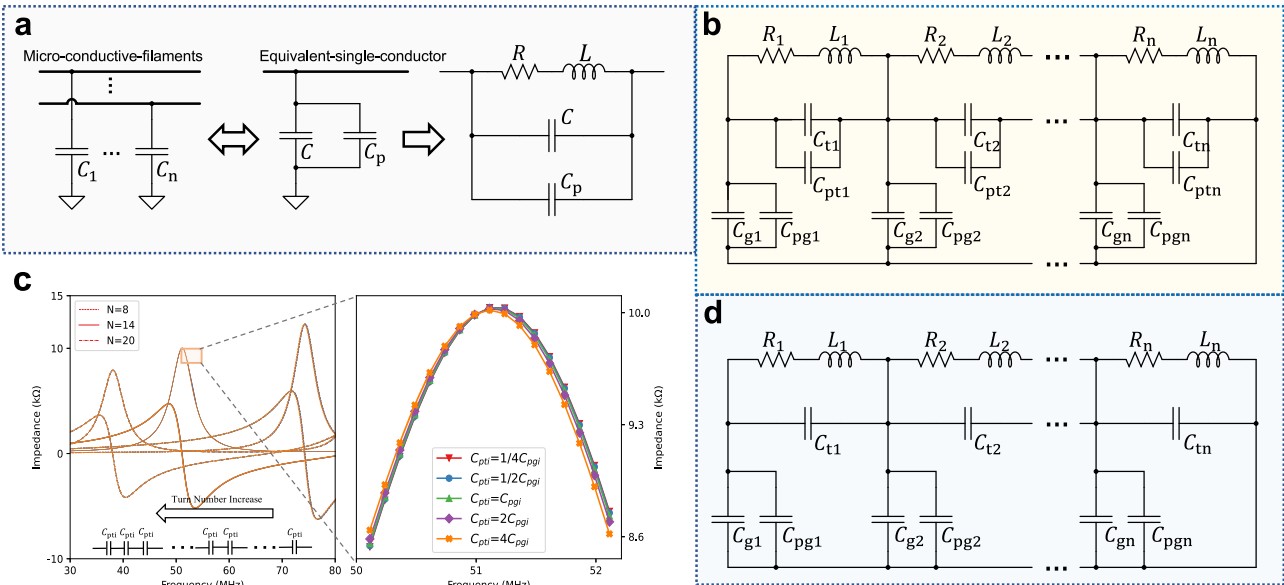

**Fig. 3 | Lump-sum and turn-to-turn model discussion. a** The total microstructure induced parasitic capacitance $C_p$ is connected in parallel to the lump-sum structural capacitance $C$. **b** Turn-to-turn model of helical inductor constructed from conductive yarn. **c** Impact of $C_{pti}$ on turn-to-turn parasitic capacitance model under different numbers of turns. **d** Simplified turn-to-turn circuit model of helical inductors constructed from conductive yarns.

as well as self and mutual inductance. A turn-to-turn equivalent circuit model of an ideal air-core helical inductor is introduced in[29], also illustrated in Fig. 2c.

In comparison with the lump-sum model, the turn-to-turn model considers the resistance of each turn segment $R_{ti}$, the self-inductance of each turn $L_{ti}$, and the mutual inductance between turns $M_{i,i+1}$, as well as the inter- and intra-turn coupling capacitances $C_{t(i,j)}$, where $i$ and $j$ represent different turns. Here only the capacitance between adjacent turns, i.e., are i.e., $C_{t(i,i+1)}$ and $C_{t(i+1,i+2)}$ are considered, and the capacitance between turns of larger separation can be ignored. We assume that when the distance between the turns is >10 times the diameter of the yarn, it is large enough to be ignored in our calculations. This model also considers the turn-to-ground capacitance $C_{gi}$, where it is the capacitance between each turn and ground. In the helical structure, the center of the coil is actually the virtual ground, and there exist voltage potentials on each turn. Therefore, the turn-to-ground capacitance can be viewed as the charges with higher potentials relative to the virtual ground on each turn.

**Models of helical inductor constructed from conductive yarns**
Based on the circuit models of helical inductor constructed from ideal metallic wires, we propose both lump-sum and turn-to-turn circuit models of helical inductor constructed from conductive yarns. The models put the parasitic capacitance (induced from the microstructure) of conductive yarns into consideration, as shown in Fig. 3a and Fig. 3b respectively.

The microstructure-induced capacitances $(C_1, \ldots, C_n)$ are formed by the filament-to-filament plating structure, while the filaments are topologically parallel to each other, the filament-induced capacitance can be regarded to be parallel-connected to the lump-sum capacitance induced from the helical structure. As illustrated in Fig. 3a, the total conductive-yarn-induced parasitic capacitance $C_p$ is in parallel with the structural capacitance $C$.

The turn-to-turn model is depicted in Fig. 3b, for a helical inductor with $N$ turns, $R_i$ $(i=1,2,\ldots,N)$ and $L_i$ $(i=1,2,\ldots,N)$ are the wire resistance and the self-inductance of the $i$-th turn respectively. $C_{ti}$ $(i=1,2,\ldots,N)$ represents structural turn-to-turn capacitances and $C_{gi}$ $(i=1,2,\ldots,N)$ represents structural turn-to-

ground capacitance as in the case of an inductor constructed from ideal metallic wires. Each turn in the helical structure is identical to each other, therefore, the value of these parameters is the same between turns, for example $(R_1=R_2=\ldots=R_i)$. The parasitic capacitance $C_{pi}$ that is produced by the microstructure of the yarn can affect both the turn-to-turn capacitance $C_{ti}$ and the turn-to-ground capacitance $C_{gi}$. To account for this, we can consider the microstructure-based capacitances $C_{pti}$ and $C_{pgi}$ in parallel with the structural turn-to-turn capacitance $C_{ti}$ and turn-to-ground capacitance $C_{gi}$, respectively, as illustrated in Fig. 3b.

From simulation, we have found out that $C_{pti}$ actually has a trivial contribution to the overall oscillating frequency of the helical inductor. A set of simulations using MATLAB Simulink® has been designed to investigate the contribution of $C_{pti}$ and $C_{pgi}$ respectively. All parameters are held constant except for $C_{pti}$. We change the relative values of $C_{pti}$ from ¼$C_{pgi}$, 1/2$C_{pgi}$, $C_{pgi}$, 2$C_{pgi}$, and 4$C_{pgi}$ respectively under different numbers of turns ($N=8, 14, 20$). The results of the real and imaginary part of the impedance are illustrated in Fig. 3c. The results demonstrate that as the relative values of $C_{pti}$ changes, the resonant frequency does not show any obvious changes. The resonant frequency only changes insignificantly. The contribution of $C_{pti}$ becomes more minute as the number of turns increases. This phenomenon can be explained from the equivalent circuit model in the bottom of Fig. 3c, as the number of turns increases, capacitances $C_{pti}$ of different turns are connected in series of each other, more turns will cause more $C_{pti}$ to be series-connected, and their overall contribution will be minimized. In fact, when the turn number is over 10, the contribution of $C_{pti}$ can be ignored. This conclusion remains valid when $C_{pti}$ is connected in parallel with $R_i$. In our experiments, the number of turns of the helical inductor ranges from 8 to 20, therefore, we can ignore $C_{pti}$ from the turn-to-turn model and only use $C_{pgi}$ to indicate the extra parasitic capacitance induced from the microstructure of the conductive yarns. In addition, although we believe the parasitic capacitance $C_{pgi}$ is originated from the gaps between the filaments in the conductive yarn, any other possible parasitic capacitances that may arise can also be included in $C_{pgi}$. The simplified turn-to-turn model is illustrated in Fig. 3d. As can be seen later, this simplification will

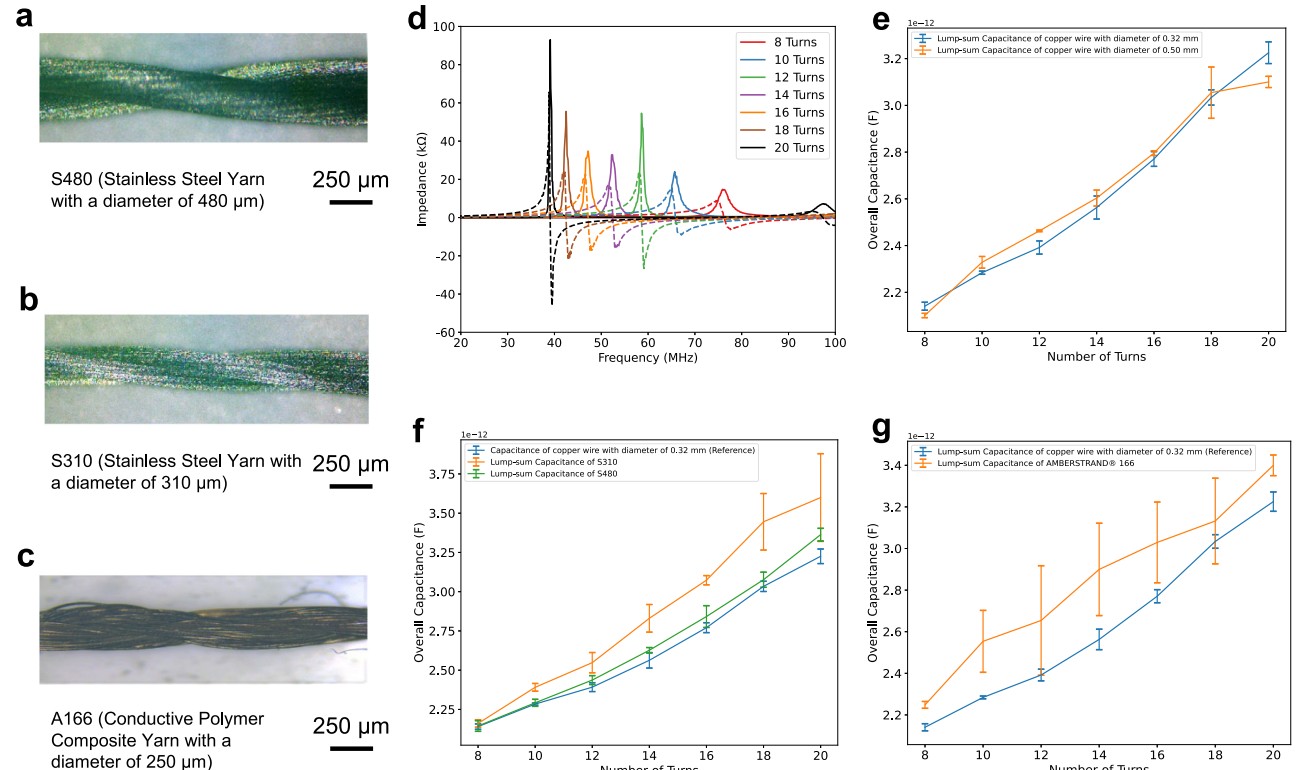

**Fig. 4 | Tested conductive yarns and measurement data.** Optical microscopic images of three kinds of conductive yarns: (**a**) S480, (**b**) S310, and (**c**) AMBER-STRAND® 166. **d** Real (solid line) and imaginary (dashed line) part of the impedance of the copper wire (0.32 mm) based DUTs with different number of turns. Lump-sum capacitance of the helical inductors made from (**e**) copper wires, (**f**) PECM conductive yarns (0.32 mm copper wire as reference), and (**g**) CPC conductive yarns (0.32 mm copper wire as reference).

help to reduce the search space in the parasitic extraction process. The value of $R_i$ and $L_i$ can be roughly estimated by following equations:

$$L_i = \frac{L}{N} \tag{2}$$

$$R_i = \frac{R}{N} \tag{3}$$

Where $L$ and $R$ are lump-sum inductance and resistance of the helical inductor as depicted in the lump-sum model. $N$ is the total number of turns.

**Conductive yarns used for parasitic capacitance estimation**

Three brands of conductive yarns are used in the experiments to extract the parasitic capacitances. S310 and S480 are PECM yarns that consist of multiple stainless-steel filaments twisted together, S310 has a yarn diameter of 0.31 mm while S480 has 0.48 mm. AMBERSTRAND® 166 is a CPC yarn that consists of Copper/Nickel/Silver layer coated on PBO Poly (p-phenylene-2,6 benzo-bisoxazole) fibers; the yarn has a high conductivity along with good mechanical properties. Pictures of these three conductive yarns are shown in Fig. 4a–c.

We also use copper wires with gauges (diameters) close to that of these three yarns to construct the helical inductors as the comparison device-under-tests (DUTs). Two sets of copper wires are used, where wire with 0.32 mm gauge is used to compare with S310 and AMBER-STRAND® 166, and 0.50 mm gauge copper wire is used to compare with S480. The gauges of the copper wires are close enough to their yarn counterparts, so the structural capacitances are similar between two helical inductors made from copper wires and yarns respectively.

**Lump-sum parameters extraction from copper wire and conductive yarn based DUTs**

The measurements of lump-sum $L$ and $R$, as well as the resonant frequencies of the helical inductors made from 0.32 mm and 0.50 mm copper wires and three conductive yarns, i.e., S310, S480 and AMBERSTRAND® 166 are illustrated in Table 2. In the table, the resistances $R$ and inductances $L$ are measured at low frequency (100 kHz, minimum operation frequency of the VNA). The resonant frequencies $f_r$ are measured at the point where the reactance is equal to 0. A testing frequency range of 100 kHz to 100 MHz is chosen because the resonant frequency of the system (created by the combination of the helical inductor and the conductive yarn parasitic capacitor) falls within this range. It should be noticed that the application frequency may differ from the test frequency used in our experiment. The results are the average of three repeated measurements. From the table, we can see that $R$ and $L$ are proportional to the number of turns $N$, as expected from the lump-sum model.

Figure 4d shows the measured impedance of the 0.32 mm copper wire based DUTs for different number of turns. The curves represent real and imaginary part of the impedance. The hyperbola shaped curve is the imaginary part and the other one is the real part. The trend of the change in amplitude of the curves is irrelevant to the study because the amplitude is influenced by the resistance of the device, which determines the Q-factor and this factor does not impact the desired results of the experiment. Using the results from Table 2, the lump-sum capacitances can be calculated from Eq. (1). The results are illustrated in Fig. 4e, f. The Fig.s show that the lump-sum structural capacitance increases almost linearly with the increase of helical turns. It also shows the copper wire gauge had little effect on the lump-sum structural capacitances, as the results from 0.32 mm copper wire are similar

**Table 2 | Lump-sum parameter measurements of helical inductors made from copper wire and conductive yarn, where C320 refers to "copper wire with diameter of 0.32 mm", and C500 refers to "copper wire with diameter of 0.5 mm"**

| Conductive trace | Turn number $N$ | $N = 8$ | $N = 10$ | $N = 12$ | $N = 14$ | $N = 16$ | $N = 18$ | $N = 20$ |
|---|---|---|---|---|---|---|---|---|
| C320 | $R$ (Ω) | 0.53 | 0.59 | 0.69 | 0.80 | 0.85 | 0.93 | 1.04 |
| C320 | $L$ (µH) | 2.04 | 2.57 | 3.09 | 3.60 | 4.12 | 4.63 | 5.15 |
| C320 | $f_r$ (MHz) | 76.12 | 65.74 | 58.49 | 52.36 | 47.11 | 42.36 | 38.98 |
| C500 | $R$ (Ω) | 0.39 | 0.45 | 0.49 | 0.51 | 0.57 | 0.61 | 0.69 |
| C500 | $L$ (µH) | 2.05 | 2.55 | 3.05 | 3.56 | 4.07 | 4.57 | 5.07 |
| C500 | $f_r$ (MHz) | 76.49 | 65.37 | 58.11 | 52.36 | 47.24 | 43.11 | 40.11 |
| S310 | $R$ (Ω) | 26.89 | 33.76 | 40.27 | 47.15 | 53.40 | 60.04 | 67.23 |
| S310 | $L$ (µH) | 2.05 | 2.56 | 3.05 | 3.52 | 4.01 | 4.51 | 4.98 |
| S310 | $f_r$ (MHz) | 75.87 | 65.74 | 58.23 | 52.36 | 47.11 | 42.73 | 38.86 |
| S480 | $R$ (Ω) | 14.99 | 18.62 | 22.35 | 26.11 | 29.62 | 33.47 | 36.93 |
| S480 | $L$ (µH) | 2.05 | 2.56 | 3.05 | 3.52 | 4.01 | 4.51 | 4.98 |
| S480 | $f_r$ (MHz) | 75.87 | 65.74 | 58.23 | 52.36 | 47.11 | 42.73 | 38.86 |
| AMBERSTRAND® 166 | $R$ (Ω) | 1.38 | 1.60 | 1.96 | 2.11 | 2.54 | 3.01 | 3.23 |
| AMBERSTRAND® 166 | $L$ (µH) | 2.21 | 2.55 | 3.12 | 3.37 | 4.04 | 4.79 | 5.15 |
| AMBERSTRAND® 166 | $f_r$ (MHz) | 71.49 | 61.49 | 55.36 | 49.98 | 45.11 | 41.23 | 38.11 |

(within measurement margin) from those of 0.50 mm copper wire. We can actually use 0.32 mm copper wire as the comparison reference to extract the parasitic capacitances in the subsequent experiments. The lump-sum capacitances of these three yarns are also extracted with the same procedure.

Using these measurements, the lump-sum parasitic capacitance of these three yarns can be extracted from Eqs. (4–6). Calculations results are listed in Table 3.

$$C_{copper} = \frac{1}{L_{copper}\left[\left(2\pi f_{r,copper}\right)^2 + \left(\frac{R_{copper}}{L_{copper}}\right)^2\right]} \quad (4)$$

$$C_{yarn} = \frac{1}{L_{yarn}\left[\left(2\pi f_{r,yarn}\right)^2 + \left(\frac{R_{yarn}}{L_{yarn}}\right)^2\right]} \quad (5)$$

$$C_p = C_{yarn} - C_{copper} \quad (6)$$

From these results, it is interesting to notice that the lump-sum parasitic capacitances of the three yarns are different to each other. While the CPC yarn (AMBERSTRAND® 166) has the largest parasitic capacitance, S480 has the smallest. These differences can be attributed to the micro-structure differences between these three wires. As seen from the magnified picture in Fig. 4c, S480 has the densest bundle of twisted filaments. From our analysis, its microstructure will create fewer number of gaps in the yarn cross-section and result in smaller parasitic capacitances.

**Table 3 | Overall parasitic capacitance in conductive yarns for different turn numbers**

| N | 8 | 10 | 12 | 14 | 16 | 18 | 20 |
|---|---|---|---|---|---|---|---|
| $C^p_{S310}$ | 0.199 | 1.066 | 1.561 | 2.674 | 3.023 | 4.113 | 3.742 |
| $C^p_{S480}$ | 0.039 | 0.081 | 0.451 | 0.641 | 0.709 | 0.424 | 1.379 |
| $C^p_{A166}$ | 1.077 | 2.696 | 2.633 | 3.366 | 2.586 | 0.983 | 1.745 |

(Unit: 0.1 pF).

## Turn-to-turn parasitic capacitance of conductive yarns

Much more complicated than the lump-sum model, in the turn-to-turn model, structural capacitances are distributed and interconnected with other turn-to-turn elements such as $L_i$ and $R_i$, and both the turn-to-turn capacitance and turn-to-ground capacitance are unknown. These variables cannot be determined simply by measuring the resonant frequencies. To summarize, the lump-sum model is used to analyze the overall parasitic capacitance of the helical inductor device, while the turn-to-turn model is used to study the parasitic capacitance of the conductive yarn on a unit length within the helical inductor. Lump-sum model is not a simple summation of the distributed model, as the lump-sum model does not take into account the distributed parasitic capacitance inside the helical coil structure. The only element that are commonly used by the two models is the inductance, where the turn-to-turn inductance $L_i$ can be calculated from the inductance of the lump-sum model using Eq. (2). It is important to note that Eqs. 4–6 are only applicable to the lump-sum model and cannot be used for the turn-to-turn model. Similarly, the values in Table 2 pertain to the lump-sum model and cannot be applied to the turn-to-turn model.

Therefore, we propose a multi-variable nonlinear regression method to determine these variables extracted from the impedance frequency response measurements at different numbers of turns. The extraction method involves two stages, i.e., Stage One and Stage Two, as illustrated in Fig. 5 and will be explained in detail in the Methods Section. In order to reduce the search space and expedite the search process, Genetic Algorithm (GA) is used.

The parameters from the turn-to-turn model can be extracted through steps illustrated in Fig. 5b. We use the non-linear regression method to extract the parameters that best fit with the measurement results from experiments. The impedance-frequency response curves of the helical inductors made from copper wires, as well as three conductive yarns are measured from the DUTs in the experiment. The simulated results from Simulink based on the extracted parameters are illustrated in Fig. 6a–d. We first extract the structural parameters using helical inductors made from 0.32 mm copper wire (Stage I results illustrated in Fig. 5a), then the turn-to-turn parasitic capacitances can be extracted in Stage II, illustrated in Fig. 5b. The extracted parameters are summarized in Table 4.

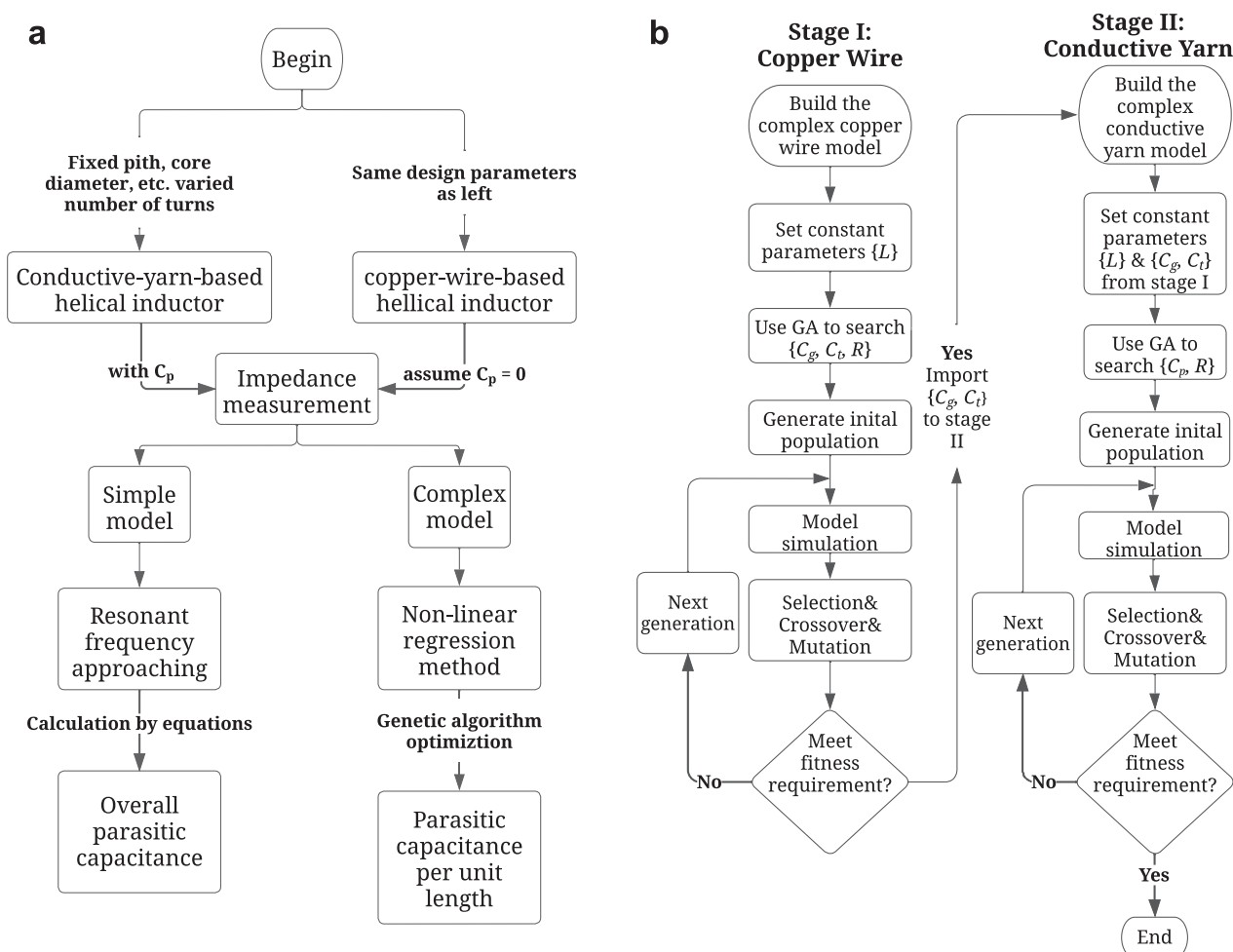

**Fig. 5 | Flowchart of optimization process. a** Non-linear regression method to extract the parasitic capacitance of conductive yarns. **b** Flowchart of the genetic algorithm with two stages.

From Table 4, the unit-length parasitic capacitance ($C_{up}$) can be derived from the parasitic capacitance of each turn:

$$C_{up} = \frac{C_p}{L_{turn}} \tag{7}$$

where $L_{turn}$ is the length of each turn, where $L_{turn} = 4\int_0^a \sqrt{1 + \frac{b^2 x^2}{a^2(a^2 - x^2)}} dx = 12.18\,cm$ in our case. $C_{up}$ for S310 is 2.71 fF/cm, while for S480 is 0.99 fF/cm, and for AMBERSTRAND 166, it has the largest $C_{up}$ of 3.53 fF/cm. These results are consistent with the lump-sum parasitic capacitance measurements of these three yarns derived from the lump-sum model in Table 3, where S480 conductive yarn has the smallest parasitic capacitance and AMBERSTRAND® 166 has the largest. It is worthwhile to mention that from the turn-to-turn model, we can see that the lump-sum capacitance is not simply the sum of turn-to-turn capacitance. Nevertheless, the relative comparison of the parasitic capacitances between these three yarns are consistent between the lump-sum model and the turn-to-turn model.

**Designing NFC coils with conductive yarn**

As a case study that incorporates the estimated parasitic capacitances of conductive yarns, we will calibrate the resonante frequency of embroidered NFC coils that are commonly used in smart apparel applications. Yarn-based NFC devices are getting increasingly popular in wearable electroncs in recent years as they benefit from the flexibility and breathability[30,31] of the fabric-based structures. Traditional inductance calculation method for NFC coils uses Wheeler's equation, which is based on the assumption of metallic conductors. NFC operation relies on the inductive-coupling between the reader coil and the tag coil, shifting of resonant frequency may have significant impact on the performance of an NFC system, especially its sensitivity and reading range. Thus, as we have demonstrated previously, the use of conductive yarns may lead to deviations in the desired inductance induced by the existence of additional parasitic capacitances.

Actually, two factors determine the extent of impact of parasitic capacitance on the performance of e-Textile devices in high-frequency applications, i.e., the operating frequency and the Q-value. On one hand, high frequency requires smaller capacitance in LC resonant circuits, where a small amount of capacitance variation will cause significant frequency shift. On the other hand, higher Q value creates narrow band-width; a small frequency shift from the peak (S11, for example) will cause significant power loss.

To evaluate this effect, several planar spiral coils with varying number of turns (5, 6, 7, 8) were designed and fabricated using AMBERSTRAND® 166 conductive yarns. The detailed design parameters can be found in the supplementary materials (Supplementary Fig. 2 and Supplementary Table 4). The inductance of each coil was calculated using Wheeler's equation.

To emulate the inductive-coupling property of an NFC device, we created a simulated NFC tag by connecting a fixed capacitor, electrically representing an NFC chip, to the embroidered antenna coils.

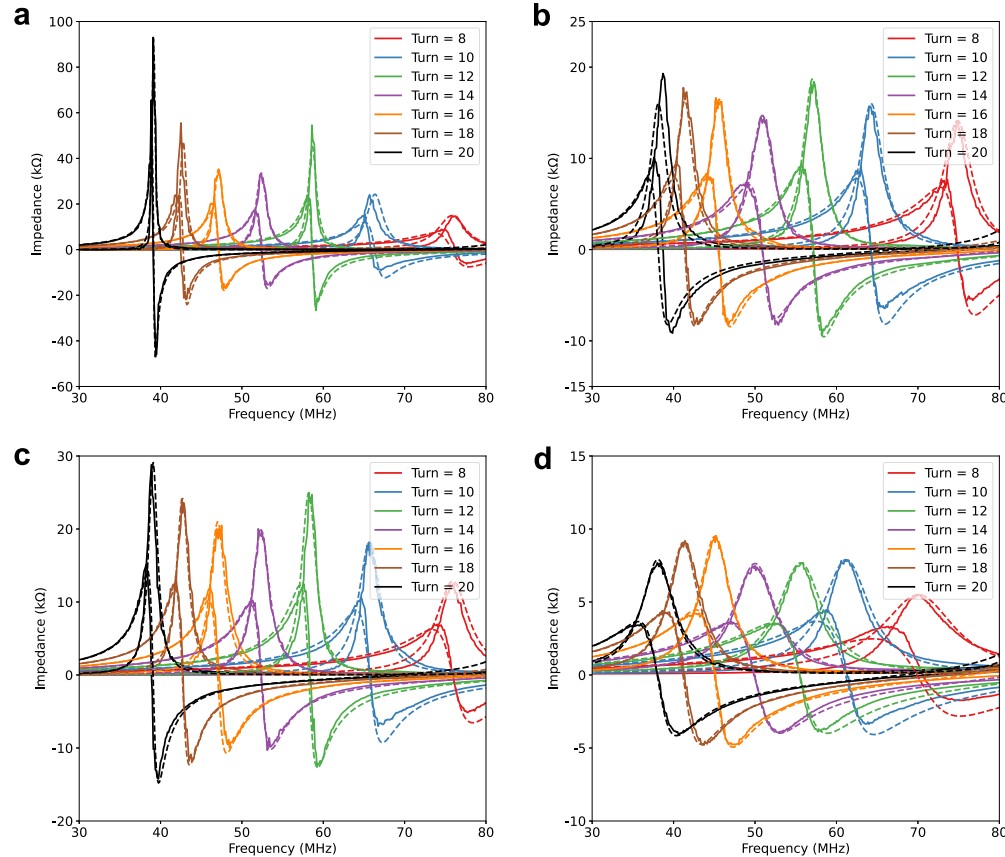

**Fig. 6 | Measurement (solid lines) V.S. Simulation (dash lines) results of the real and imaginary part of the impedance. a** copper wire (0.32 mm) based DUTs, **b** S310 based DUTs, **c** S480 based DUTs, and **d** AMBERSTRAND® 166 based DUTs.

The calculated inductance (see Methods Section) of the coils and the capacitance of the external capacitor were used to calculate the theoretical optimal operating frequency, which is given by the equation $f = \frac{1}{2\pi\sqrt{LC}}$. We then measured the actual resonant frequency of the tag using an NFC testing platform (QBG5C, AI YI).

Supplementary Table 5 presents a comparison of the theoretical and actual resonant frequencies of coils with various turn numbers (5–8), using a 25 pF capacitor. The calculated frequencies were 25.431, 21.192, 18.320, 16.025 MHz respectively and actual measured frequencies were 25.205, 21.067, 18.131, 16.009 MHz respectively. The results show deviations of −0.226, −0.125, −0.189, −0.016 MHz respectively between the calculated and measured values. It is anticipated that the actual frequency is slightly lower than the calculated one due to the fact that Wheeler's equation only considers structural parasitic capacitance, but not the parasitic capacitance of the yarns. This additional capacitance is expected to lower the actual resonant frequency.

We then estimated the yarn parasitic capacitance for each coil by multiplying the yarn's total length and the extracted unit length parasitic capacitance of AMBERSTRAND® 166. This allowed us to

compute an amended frequency using the equation $f = \frac{1}{2\pi\sqrt{L(C+C_p)}}$, where $C_p$ is the calculated yarn parasitic capacitance. For instance, the coil with 6 turns has a length of 77.28 cm, generating a calculated paracitic capacitance of 0.273 pF (77.28 × 3.53 = 273 fF). The comparison of the amended frequency and measured frequency is presented in Supplementary Table 6. In comparison to Supplementary Table 5, the deviation has been reduced and the actual frequency is now much closer to the actual measured frequency, with an average deviation of only around 0.01 MHz.

## Discussion

In summary, we proposed methods and techniques to extract the parasitic capacitances induced from the micro-structures (twisted filament threads) of conductive yarns. Our research attempts to comprehensively address, model, and quantify the parasitic capacitances of conductive yarns, which are of great theoretical importance. We have estimated that the parasitic capacitance is in the range of femtofarad (fF); this estimation can be used to elucidate and compensate design variations in higher-frequency and RF applications. The proposed methodology and circuit models, along with measurement results can be used to accurately simulate the performance of e-Textile devices.

## Methods

### Parasitic capacitance extraction methods

In order to extract the conductive yarn's parasitic capacitance $C_p$ in the lump-sum model, as well as the capacitance $C_{pi}$ of each turn in the turn-to-turn model. We construct two structurally identical air-core helical inductors; one is wound from conductive yarns, while the other is wound from copper wires (as approximate alternative to ideal metallic

**Table 4 | Parameters for each turn of simplified turn-to-turn circuit model of helical inductors (Fig. 3d)**

| | $R_i^*$ ($\Omega$) | $L_i$ ($\mu$H) | $C_{ti}$ (pF) | $C_{gi}$ (pF) | $C_{pgi}$ (pF) |
|---|---|---|---|---|---|
| Copper Wire (Reference) | 2.81 | 0.26 | 7.500 | 0.333 | 0 |
| S310 | 7.14 | 0.24 | 7.500 | 0.333 | 0.033 |
| S480 | 4.28 | 0.25 | 7.500 | 0.333 | 0.012 |
| AMBERSTRAND® 166 | 10.14 | 0.26 | 7.500 | 0.333 | 0.043 |

*The resistance of each turn varies with the number of turns (which affects the Q factor), here we use the average value.

wires) of similar gauges. Similar gauge means that the diameter of the copper wires is close to that of the conductive yarns, therefore the intrinsic structural capacitances, i.e., the turn-to-turn capacitance and turn-to-ground capacitance are almost identical. By comparing the resonant frequency and impedance frequency response curve between these two inductors, parasitic capacitance induced from the microstructure of the conductive yarns can be extracted.

We propose two methods to extract these parasitic capacitances, i.e., lump-sum parasitic capacitance can be extracted by measuring the resonant frequency differences (shifting), while the turn-to-turn parasitic capacitance can be extracted by multivariable nonlinear regression techniques applied on the impedance frequency responses. The flowchart of these two methods is illustrated in Fig. 5.

### Calculating the lump-sum inductance $L$ and Resistance $R$

Based on the lump-sum equivalent circuit derived in Fig. 2a, the resonant frequency of a helical inductor can be calculated using Eq. (1). Specifically in our experiments, $f_r$ can be derived from the impedance sweeping on a network analyzer, when the imaginary part of the impedance is equal to 0. For the helical inductor with a sufficiently higher inductance, under low frequency, the reactive value of the inductor is mainly determined by the inductor $L$, while the structural capacitance can be ignored; therefore, the lump-sum $L$ can be measured at sufficiently low frequency (in our case, a few Kilo-Hertz). The lump-sum capacitance $C$ can be derived from the resonant frequency $f_r$ and the lump-sum inductance $L$ using Eq. (1). The lump-sum resistance can be directly measured using a four-point-probe resistivity measurement apparatus. For example, to measure the lump-sum resistance, we applied a DC current ($I$) between the outer two probes and then measured the voltage drop ($\Delta V$) between the inner two probes. We calibrated the measurement to account for contact resistance. The resistance was calculated by dividing the voltage drop ($\Delta V$) by the current ($I$).

### Extracting the lump-sum parasitic capacitance

The above method can be used to determine the structural lump-sum capacitance of the copper-wound inductors $C_{copper}$ and the yarn-wound inductors $C_{yarn}$. Based on the lump-sum model, the difference between $C_{copper}$ and $C_{yarn}$ is the lump-sum parasitic capacitance induced from the yarn's microstructure. The calculation can be formulated through Equation (4) to Eq. (6).

### Extracting the turn-to-turn parasitic capacitance

The solution space vector of the unknown variables in the turn-to-turn model is ($R_i$, $L_i$, $C_s$, $C_c$, $C_p$). $L_i$ and $R_i$ can be first calculated by Eq. (3) and Eq. (4). Therefore the solution vector is reduced to ($\frac{R_t^{copper}}{N}$, $\frac{L_t^{copper}}{N}$, $C_s$, $C_c$, $C_p$), where $R_t^{yarn}$ and $L_t^{yarn}$ are total resistance and total inductance of the copper inductor measured at low frequency; N is the number of turns.

The goal of Stage One is to first determine $C_s$ and $C_c$, in order to reduce the search space, $C_p$ is set to be 0 at this stage. The solution vector can be further reduced to ($\frac{R_t^{copper}}{N}$, $\frac{L_t^{copper}}{N}$, $C_s$, $C_c$, 0). The cost function to be minimized in the search process is defined as follows in Eq. (8):

$$F = \sum_{i=1}^{M} \left\{ |Re\{Z_{meas,i}\} - Re\{Z_{sim,i}\}|^2 + |Im\{Z_{sim,i}\} - Im\{Z_{sim,i}\}|^2 \right\} \quad (8)$$

where $M$ is the total number of sample points of the measurements, $Z_{meas}$ is the impedance frequency response measured from the network analyzer, and $Z_{sim}$ is the simulated impedance frequency response from Simulink. The search is to minimize the cost function $F$. GA is used to expedite the search in Simulink. GA's search rules are summarized in Supplementary Table 1. Because the copper inductor and the yarn-based inductor have identical helical structure, $C_s$ and $C_c$

values should be the same. Therefore, we use the measurements of copper indicator as the search target, which is denoted as ($\frac{R_t^{copper}}{N}$, $\frac{L_t^{copper}}{N}$, $C_s^{copper}$, $C_c^{copper}$, 0).

For Stage Two, the impedance measurements from yarn-based inductors are used as targets in this stage. At this stage, the solution vector is reduced to ($\frac{R_t^{yarn}}{N}$, $\frac{L_t^{yarn}}{N}$, $C_s^{copper}$, $C_c^{copper}$, $C_p$), where $R_t^{yarn}$ and $L_t^{yarn}$ are total resistance and total inductance of the yarn-based inductor measured at low frequency; $N$ is the number of turns; $C_s^{copper}$ and $C_c^{copper}$ are determined from Stage One. The search process is summarized in Supplementary Table 2, along with the search ranges of each variable used in our experiments.

### Helical inductors construction and ipedance measurement

As discussed previously, two identically structured helical inductors, one constructed from copper wires, and the other from conductive yarns, are constructed and used as the devices-under-test (DUTs) for the impedance measurement.

In order to accurately extract the parasitic capacitance from the yarns, the structural capacitance cannot be overwhelmingly larger than the parasitic capacitance; otherwise the measurements will be dominated by the structural capacitance. The helical inductor is constructed with a larger spacing (10 mm) between the turns as well as a larger diameter (44 mm) of each turn (the parameters are provided in Supplementary Table 3, as shown in Supplementary Fig. 1b). Helical structures of different numbers of turns (from 8 to 20 with an increment of 2) are also constructed as needed by the nonlinear regression extraction method.

The air-core helical structure is actually supported by an ABS plastic cylinder (thickness of 2.5 mm) for both copper and yarn inductors. The copper wires and yarns are fit to the grooves that are pre-carved on the cylinder; such the helical structure is stable during the repeated measurements. Overall structure of the helical inductor is shown in Supplementary Fig. 1a.

Keysight® Vector Network Analyzer (VNA) E5071C was used in the experiment. Because of the spacing and turn-diameter of the helical structure, the DUT's overall length ranges from 74 mm (8 turns) to 200 mm (20 turns), the measurement cannot be performed with a single-ended SMA port. Port extension apparatus is also designed. The extension used in our experiments consists of a jumper wire, two wire clamps, and an SMA connector. One end of the jumper wire is soldered onto the SMA connector, while the other end is connected to one of the wire clamps. To maintain consistency in our data, we set the distance between the ends of the helical inductors and the extension to be equal to the length of half a turn. Additionally, we included the extension in the calibration process (using the Port Extension mode on the VNA) to minimize its impact on the measurements. Detailed structure of extension is shown in Supplementary Fig. 1b and Supplementary Fig. 1c, illustrating the measurement setup of the VNA.

Since both the inductor and VNA are sensitive to ambient interference, the measurement is performed in an anechoic chamber as depicted in Supplementary Fig. 1c. The impedance of the DUT inductors is measured under the frequency sweeping from 100 kHz to 100 MHz. This range has accommodated all first resonant frequencies of different turns (from 8 to 20) used in this experiment. VNA as well as the extension apparatus are calibrated using Keysight® ECal Module N7552A.

### Inductance calculation using Wheeler's equation

The inductance of each coil was calculated using following equation[18]:

$$L_{circle} = 31.33\mu_0 N^2 \frac{a^2}{8a + 11c} = 31.33\mu_0 N^2 \frac{a}{8 + 22\rho} \quad (9)$$

In this equation, $\mu_0$ represents the magnetic permeability of free space ($\mu_0 = 4\pi \times 10^{-7}\,Hm^{-1}$), $N$ is the number of turns in the coil, $a$ is

the average radius of the coil ($a = \frac{d_{out} + d_{in}}{4}$), $c$ is the distance between the inner turn and outer turn, and $\rho$ is defined as $\rho = \frac{d_{out} - d_{in}}{d_{out} + d_{in}}$.

## Fabrication of NFC coils

The commercial software PE-DESIGN 10 was used to convert the digitalized stitch trajectory of the coil pattern. We fabricated several prototypes of the planar spiral coils in the shape of circular using the digital embroidery process, as shown in Supplementary Fig. 3a. The digital embroidery machine PR670E, along with the conductive yarns AMBERSTRAND® 166, which served as the bottom bobbin to support the upper textile thread, used in this process. A consistent stitch size of 2 mm is used to fabricate the coil across all settings. The spiral coil prototypes were then embroidered on a flexible cotton substrate, as shown in Supplementary Fig. 3b.

## Measurement of NFC resonant frequency

After the e-textile coils were fabricated using conductive yarns, we investigated their radio frequency (RF) properties. By connecting a capacitor to the terminals of the embroidered coils, we were able to create an NFC resonant tag. We used the Wheeler's equations to estimate the inductance ($L$) of planar spiral coil inductors with different geometrical parameters. With the calculated $L$ of the inductors and the selected capacitance ($C$) values, we were able to determine the theoretical optimal operating frequency using the equation $f = \frac{1}{2\pi\sqrt{LC}}$, as listed in Supplementary Table 5–8.

To measure the self-resonance frequency of the embroidered coils with capacitors, we used the HF/LF Tag Test machine (QBG5C, AI YI). We connected capacitors to the terminals of several prototypes to account for individual variations. Once the tag devices were placed steadily on the test platform, we were able to read the resonant frequency value directly from the platform, As shown in Supplementary Fig. 3c.

## Data availability

The data that support the findings of this study are available within the main text and Supplementary Information. Source data are provided with this paper. Figshare https://doi.org/10.6084/m9.figshare.22574179 Source data are provided with this paper.

## Code availability

The code that support the findings of this study are available from the corresponding author upon request.

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

## Acknowledgements

We would like to acknowledge the financial support provided by the Key-Area Research and Development Program of Guangdong Province (Grant No. 2020B0101030002) and the Shenzhen Science and Technology Program (Grant No. JCYJ20190809115803580) for the successful completion of our research work.

## Author contributions

T.Y. conceived of this project. Z.Q., Z.Z. and Y.L. contributed equally to this work, should be considered as co-first author. Specifically, Z.Q. and Z.Z. prepared and conducted the experiments. Y.L. designed the experiments. M.Y. prepared planar spiral coils and conducted NFC resonant frequency tests. Draft was written by Z.Q., Z.Z and Y.L, and revised by T.Y. All authors discussed and approved the results of this paper.

## Competing interests

The authors declare no competing interests.
