## [Peer Review File · Nature Communications]

Parasitic Capacitance Modeling and Measurements of Conductive Yarns for e-Textile DevicesREVIEWER COMMENTS

Reviewer #1 (Remarks to the Author):

This paper presents a mathematical model of the parasitic capacitance, which should contribute to subsequent work of e-textiles. The waves are similar to those of real data, and the possibilities are promising. The results of trials on several commercial materials are useful. However, there are several core points that are not clear, and if these are not resolved, the reproducibility is doubtful. I would recommend that we review the paper again after major revision.

First, an important previous paper is missing. In [A], a helical structure around an insulator is realized, and there is even an introduction of a conducting core, which is not in the proposed method. It is an exaggeration to say that the authors are the first to propose an air-core helical inductor model, and they should properly revise their position.

[A] Tajitsu, Y. et al. New wearable sensor in the shape of a braided cord (Kumihimo). IEEE Transactions on Dielectrics and Electrical Insulation 25, 772-777 (2018).

I think that there is an assumption that air-core does not bend, but in fact that assumption is not quite correct. For example, Figure 1(c) shows a situation where the air-core seems to be bent, and [A] shows it is braided. An air-core can also change its shape when an external force is applied. The deformation and capacity change when external pressure is applied is discussed in [B]. I would like to see justification for modeling the air-core assuming that it does not bend. Or, please specify it as a limitation and add discussions.

[B] Terada, T., Toyoura, M., Sato, T. & Mao, X. Functional Fabric Pattern—Examining the Case of Pressure Detection and Localization. IEEE Transactions on Industrial Electronics 66, 8224-8234 (2019).

The authors discuss impedance at frequencies of several 10 MHz. What kind of application is envisioned? It seems difficult to use it for acquiring biological signals. Please add discussions.

There is one major difference between the simulation results shown in Figure 3(c) and the measurement results shown in Figure 4(d). In the simulation, the waves become larger as the frequency increases, while they become smaller in the measurement results. I cannot determine whether this is due to the influence of the real environment or to incomplete mathematical modeling. I would appreciate an appropriate discussion.

It should be clearly stated how both ends of the yarn were treated. The statement is necessary to ensure reproducibility.

The scales of the three pictures in Figure 4(a) are different and should be the same. If there is no special reason, the backgrounds should also be the same. It is difficult to tell the difference when they are placed side by side.

There is an unnecessary closing bracket on line 131.

Reviewer #2 (Remarks to the Author):

The paper investigates a fundamental issue of parasitic capacitance which exists by nature in most applications with conductive materials. In particular, the paper is focused on the discovery of parasitic capacitance in electronic textiles sector where conductive yarns are commonly used. The paper presents two models, lump-sum and turn-to-turn models, to extract the parasitic capacitance with aid of a MATLAB modelling and experiences. Three commercially off the shelf conductive yarns have been tested in addition to their countermodels made using copper wires. Overall capacitance and unit capacitance have been derived from these methods and authors claims the models can be used to estimate parasitic capacitance in other scenario.

However, the reviewer has the following concerns regarding this manuscript:

1.The paper claims that “parasitic capacitance greatly affects the performance of e-textile device in RF and high frequency applications” but fails to explore in such frequency bands. The maximum frequency explored in the paper is 100MHz which is well below common RF bands. Since the capacitance is frequency dependent, the paper should investigate the impact of parasitic capacitance in the correct RF bands.

2.Following the Q1, how significant the parasitic capacitance is in e-textile application and what impact the parasitic capacitance will be? The paper should compare and analyse in a case study to demonstrate the importance of parasitic capacitance in e-textile sector. The significance needs to be quantified so the method can greatly benefit a wider community.

3.The model of parasitic capacitance in a conductive yarn, shown in Fig 1 (c), is questionable. Depending on the composite of yarns, there should not be any capacitance between two conductive filaments if it is made of stainless steel. Under twisting process of filament during the yarn manufacturing pushes conductive filaments together. This will at a result, create a conductive path between two filaments, assuming the filament is 100% stainless steel. This conductive path will therefore eliminate any capacitor being formed. The paper seems to assume that there is an air gap between two filaments so capacitance exists. This is not a case due to the reason explained above. Authors needs to explain more about materials with the support of any assumptions.

4.Line 263, the paper claims that the length of each helical turn is the perimeter of a circle. This is only true when the space between two turns is much smaller compared to the diameter of the turning circle. However, this is not true with the setup of this paper. The 10mm space between turns is compared to 44mm of turn diameter. The unit turn length is not equal to the perimeter of 44mm circle.

5.The measurement using VNA uses port extension with jumper wires which could introduce parasitic capacitance. How can this capacitance be minimised and then eliminated?

6.Authors need to be careful when using vague wordings, such as “very insignificant”, “even more minimal” etc. Those wordings should not be in the paper unless being backed by the scientific evidence.

7.Some other points of considers:

Line 20: what is meant by “yarn`s microstructure”?

Line 53: “high frequency RF applications” have not been defined.

Line 79: why S310, S480 and 166 are used and how can these three commercial yarns represent the parasitic capacitance of overall conductive yarns? Have you considered filament dimension in connection with the yarn dimension?

Line 114: stop using “etc” when describing the notation and meaning of symbol. Just say what are they.

Line 132: “capacitance between two turns of large separation can be ignored”. Please define how large is large and what is the assumption here.

Line 160: figure 3 (c) needs to be improved in terms of its presentation. Colouring is not a good option when viewing in black/white. Suggesting to add symbols to each curve.

Line 269: typo. Should it be “comparison” not “caparison”?

Line 306: How lump-sum resistance is measured using a four-point-probe?

Reviewer #3 (Remarks to the Author):

Very interesting topic! I myself have been thinking about the reactive properties of electrically conductive yarns. I also applaud the will to characterise these kinds of yarns in a systematic way without having a specific application in mind. I do have some questions regarding your work. I have attached a word document with some feedback and questions for you.

Very interesting topic! I myself have been thinking about the reactive properties of electrically conductive yarns. I also applaud the will to characterise these kinds of yarns in a systematic way without having a specific application in mind. Having said that, I would argue that most of the time these conductive yarns are used to form either woven, knitted or embroidered structures where many conductive yarns are interconnected. In such situations, I believe that the dominating capacitances would either be those between sets of yarns in close proximity to each other or between a set of yarns and other bodies (e.g. ground). I did very rudimentary measurements (frequency response analysis) of conductive fabrics and yarns to see if the mechanical tensile load would affect the impedance of the yarn or fabric. I swept the frequency between 10 Hz up to 10 MHz and applied different tensile loads on the samples and I could not see any change apart from the resistance of the samples. I think that as long as one does not have any significant inductance in the samples, the potential variation of the stray or structural capacitances in yarns or simple fabrics themselves have a very minute effect on the impedance.

I do not wholly understand your equivalent circuit in Figure 1 d though. Looking at Figure 1 a it seems to me that the filaments touch each other at the ends of the gaps indicated. That, to me, would suggest that there should be a contact resistance in parallel with the parasitic capacitances (see my annotated version of your Figure 1). Unless of course that each filament has an insulating layer on its surface. On line 100, you state: "Because of the intrinsic resistance of the yarn materials, adjacent filaments may not have the same potential." I could not find a reference to any data sheet about the yarns you used, it could be good if you added that on lines 81-83.

So are the filaments in contact with each other along the yarn or are they only in contact at the terminals? If they are in contact along the yarn then I would like an explanation of how you neglect the contact resistance between the filaments. If they are not then I would believe that the part C_{pti} (on line 165) would make a large contribution to the overall capacitance. In addition, if the filaments are in contact with each other then how uneven is the conductivity of the medium? Because if the length of the gaps (not the distance between the conductive surfaces) is of the order of 100 μm , then how likely is it that two filaments have different potential (i.e. is the relaxation time of the conductive medium greater than, say $1/1e^8 = 10 \text{ ns}$)? For a homogeneous piece of copper, I think that the relaxation time is of the order of $1e-18 \text{ s}$.

I would guess that these contact resistances would play a role in the overall impedance as well

Again about possible contacts between the filaments: the reasoning on lines 145 – 149 leading to the statement that the micro-structural capacitances could be viewed as connected in parallel to the turn-to-turn capacitance of the coil. If such contacts exist then perhaps the yarn could be modelled as a parallel R-C link per unit length, and if so, should not C_{pti} be in parallel with R_i rather than with the series connection of R_i and L_i (as in Figure 3b)?

On line 156 you state that the C_{pi} may contribute to C_{ti} and C_{gi} , how does it contribute to C_{tgi} ?

About Figure 3d, 4d and 6, are the graphs absolute value and phase angle of the impedance or is it the real and imaginary parts of the impedance? In either case, it could be nice to write that out in the text somewhere.

So if I understand correctly the C_{ti} does not contribute in any significant way to the overall impedance. That is also what I found when I did some SPICE simulations using first your model (Figure 3b) and the values you report (and later on your simplified model in Figure 3d). I am a little bit confused about what the different reported values represent though. Using the values of Table 2 for 8 turns of the yarn S310 straight off and in addition the value for C_{S310}^p for 8 turns in Table 3 as 0.0199pF, one would get the following situation with your simplified tur-to-turn model (Figure 3d):

And running the simulation one gets the following spectra:

I did vary the C_{pg} between 20-50 fF. Neither of those peaks resemble the ones seen in Figure 3c, 4d or 6b. So I guess I didn't interpret your text correctly.

On lines 182-184 you state that the values for each R_i and L_i in the model can be taken as the measured total R and L divided by the number of turns. So using the yarn labelled S310 with 8 turns one would, from table 2, get $R_i = 3.36 \Omega/\text{turn}$ and $L_i = 0.256 \mu H/\text{turn}$. And using Equations 4-6 I get $C_{copper} = 2.143 pF$ and $C_{yarn} = 2.145 pF$ giving $C_p = 2 fF$. Now this C_{yarn} and C_p I take it is for the entire circuit so it should also be divided by the number of turns, right? In that case one gets $C_{yarn,i} = 0.268 pF/\text{turn}$ and $C_{pi} = 0.25 fF/\text{turn}$. Then one gets the following:

And the impedance spectra as

In essence, I have difficulties reproducing your graphs. So maybe you could describe in a more clearer way. The only way I could get an impedance graph that resembles the one in Figure 6 b was if I used the values from Table 2 and the C_{S310}^p for 8 turns in Table 3 but ten times smaller in a single lumped-element model (i.e. 1.99fF instead of 19.9 fF). Like this:

.ac dec 20000 30meg 80meg
.param Cpg = 1.99f

Then I can get a peak that is situated at ca 75.8MHz, but the peak value of the real part is slightly more than two times the value of the peak in Figure 6b.

I am not familiar with the Genetic Algorithm that you use for extracting the turn-to-turn parasitic capacitances, so I do not have any input on that.

On line 258 (Table 4) you introduce C_s and C_c , what are those? Also in Table 4 you say that the R_t for the copper wire is 2.81Ω per turn, but in Table 2 you measured it to be between $0.0517 - 0.0663 \Omega/\text{turn}$ (lowest for 18 turns and highest for 8 turns). How are the values in Table 2 and Table 4 related?

On lines 83-86 you say. "It is also important to mention that, like many other parasitic parameter extraction techniques used for electronic devices. The extraction results are affected by many structural and ambient conditions and cannot be very accurate." Then on lines 198-207 you describe how you got the values for the total resistance and inductance and also how you measured the resonance frequency. These measurements are then used, if I understand correctly, to extract the total equivalent capacitance using Equation 1. The errorbars in Figure 4e for the 0.32 mm copper wire at 8 turns seems to be around 50fF top to bottom and in Table 3 the calculated value of C_{S310}^p for 8 turns is 19.9 fF. This, in addition to your statement on lines 267-269 that the extracted values for the turn-to-turn model do not add up to the lump-sum value makes me wonder if there might be some factor that is not taken into consideration in your model. Having said that, I think it is nice to see that, as you say, there is a correlation between the lump-sum extracted values and the turn-to-turn one.

It would be good if you could specify the different graphs in Figure 6, just as you did in Figures 3c and 4d. I do also recommend you to use more differentiable colours or perhaps different line types. For all figures showing the impedance spectra, it would be good if you could utilise the right hand y-axis to state if it is phase angle or imaginary part of the impedance.

REVIEWER COMMENTS

Reviewer #1 (Remarks to the Author):

This paper presents a mathematical model of the parasitic capacitance, which should contribute to subsequent work of e-textiles. The waves are similar to those of real data, and the possibilities are promising. The results of trials on several commercial materials are useful. However, there are several core points that are not clear, and if these are not resolved, the reproducibility is doubtful. I would recommend that we review the paper again after major revision.

Reply: Thank you for your review of our manuscript. We appreciate your positive feedback and believe that our mathematical model of parasitic capacitance will be useful for future research on e-textiles. We also understand your concerns about the clarity of certain aspects of the paper and the potential impact on reproducibility. We will make every effort to address these issues in the revised manuscript and provide additional data and analysis as needed. We appreciate the opportunity to revise the manuscript and hope that it will be suitable for publication upon further review. Please note that the corresponding revised content is included here and also highlighted in the revised manuscript for your convenience. Thank you again for your feedback and assistance in improving the quality of our work.

1. First, an important previous paper is missing. In [A], a helical structure around an insulator is realized, and there is even an introduction of a conducting core, which is not in the proposed method. It is an exaggeration to say that the authors are the first to propose an air-core helical inductor model, and they should properly revise their position.

[A] Tajitsu, Y. et al. New wearable sensor in the shape of a braided cord (Kumihimo). IEEE Transactions on Dielectrics and Electrical Insulation 25, 772-777 (2018).

Reply: Thank you for pointing this out. We are sorry for the use of the misleading word "first" in line 111 of the original manuscript. This word was not intended to imply that we were the first to propose the air-core helical inductor model, but rather to refer to the order in which our experiment was conducted. What we intended to convey is that "firstly, we will introduce an air-core helical inductor model." Thank you for bringing this to our attention. We have made the necessary corrections in the revised manuscript. Additionally, the paper by Tajitsu, Y. [A] offers a new perspective on this topic and will be included in Table 1.

Line 125-126 (revised manuscript):

“Firstly, we introduce the lump-sum circuit model of the helical air-core inductor constructed from ideal metallic wires”

Line 38-39 (revised manuscript):

“Table 1 lists some typical e-textile devices along with the conductive yarns used for construction [7, 20-23, 30-31].”

Line 47 (revised manuscript):

“Table 1. Conductive-Yarns-Based Devices Applications.

...
[30]	Piezoelectric Sensor	Piezoelectric poly-l-lactic acid (PLLA) fibers	Large Impedance, High Noise, Unsensitive with Specific Structures

”

Line 570-571 (revised manuscript):

[30] Tajitsu, Y. et al. New wearable sensor in the shape of a braided cord (Kumihimo). IEEE Transactions on Dielectrics and Electrical Insulation 25, 772-777 (2018).

2. I think that there is an assumption that air-core does not bend, but in fact that assumption is not quite correct. For example, Figure 1(c) shows a situation where the air-core seems to be bent, and [A] shows it is braided. An air-core can also change its shape when an external force is applied. The deformation and capacity change when external pressure is applied is discussed in [B]. I would like to see justification for modeling the air-core assuming that it does not bend. Or, please specify it as a limitation and add discussions.

Figure 1(c).

[B] Terada, T., Toyoura, M., Sato, T. & Mao, X. Functional Fabric Pattern—Examining the Case of Pressure Detection and Localization. IEEE Transactions on Industrial Electronics 66, 8224-8234 (2019).

Reply: Thank you for your suggestion and for bringing this issue to our attention. We understand the rationale behind questioning whether bending could potentially affect the air-core model. In our experiment, the air-core inductor did not bend. In Figure 1(c), we were actually referring to the cross-section of the conductive yarn rather than the air-core helical inductor. The purpose of this graph is to illustrate the source of parasitic capacitance and how it is generated. We apologize for any confusion caused by our lack of clarity in the description of Figure 1(c) in the original

manuscript. We have revised the text to address this issue in the revised manuscript. The paper by Terada, T[B] provides new insight to our work and will be included in the revised manuscript.

Line 96-97 (revised manuscript):

“Figure 1. (a) ... (c) An equivalent circuit model of parasitic capacitances formed by the gaps between adjacent filaments, and (d) Cross section of the conductive yarn: parasitic capacitances are uniformly distributed along the conductive yarns.”

Line 111-120 (revised manuscript):

“A distributed equivalent circuit model of the parasitic capacitances created by the gaps is illustrated in Figure 1(c), where R_{seg1_top} and R_{seg2_top} , etc. are the resistance of the filament segments that form the top plate of parasitic capacitances between two contact points, and R_{seg1_bottom} , R_{seg2_bottom} , etc., are the resistance of the filament segments that form the bottom plate. R_c is the contact resistance formed by the adjacent filaments touching each other at the ends of the gaps. C_{gap1} and C_{gap2} are the parasitic capacitances created by the gaps.

The capacitance formed by each tiny gap can be consolidated into parasitic capacitance between adjacent filaments. Figure 1(d) illustrates a cross section of the conductive yarn and the collective parasitic capacitances between the filaments.”

Line 47 (revised manuscript):

“Table 1. Conductive-Yarns-Based Devices Applications.

...
[31]	Pressor Sensor	Silver-plated nylon fiber coated with polyester	low sensitivity, low robustness

”

Line 572-573 (revised manuscript):

[31] Terada, T., Toyoura, M., Sato, T. & Mao, X. Functional Fabric Pattern—Examining the Case of Pressure Detection and Localization. IEEE Transactions on Industrial Electronics 66, 8224-8234 (2019).

3. The authors discuss impedance at frequencies of several 10 MHz. What kind of application is envisioned? It seems difficult to use it for acquiring biological signals. Please add discussions.

Reply: Thank you for bringing this important point to our attention. It is important to note that the application frequency may differ from the test frequency used in our experiment. However, it is worth noting that parasitic capacitance is an intrinsic property of the material that is influenced by factors such as permittivity, yarn diameter, and spacing, rather than frequency. Therefore, it is not affected by the frequency of the application.

In our experiment, we chose a testing frequency range of 100kHz to 100MHz because the resonant frequency of the system (created by the combination of the helical inductor and the conductive yarn parasitic capacitor) falls within this range. We use the resonant frequency to extract the parasitic capacitance, which is therefore independent of the application frequency.

In the revised manuscript, we will also include additional case studies of potential applications to provide more details on how the parasitic capacitance may impact these applications and why our research is relevant in these contexts. Thank you again for raising this valuable question.

Line 309-342 (revised manuscript):

“Designing NFC Coils with Conductive Yarn

As a case study that incorporates the estimated parasitic capacitances of conductive yarns, we will calibrate the resonant frequency of embroidered NFC coils that are commonly used in smart apparel applications. Yarn-based NFC devices are getting increasingly popular in wearable electronics in recent years as they benefit from the flexibility and breathability [28,29] of the fabric-based structures. Traditional inductance calculation method for NFC coils uses Wheeler's equation, which is based on the assumption of metallic conductors. As we have demonstrated previously, the use of conductive yarns may lead to deviations in the desired inductance induced by the existence of additional parasitic capacitances.

To evaluate this effect, several planar spiral coils with varying number of turns (5, 6, 7, 8) were designed and fabricated using AMBERSTRAND® 166 conductive yarns. The detailed design parameters can be found in the supplementary materials (Figure S2 and Table S4). The inductance of each coil was calculated using Wheeler's equation.

To emulate the functionality of an NFC device, we created an NFC resonant tag by connecting a capacitor to the embroidered coils. The calculated inductance (see Methods Section) of the coils and the capacitance of the external capacitor were used to calculate the theoretical optimal operating frequency, which is given by the equation $f = \frac{1}{2\pi\sqrt{LC}}$. We then measured the actual resonant frequency of the tag using an NFC testing platform.

Table S5 presents a comparison of the theoretical and actual resonant frequencies of coils with various turn numbers (5-8), using a 25 pF capacitor. The calculated frequencies were 25.431, 21.192, 18.320, 16.025 MHz respectively and actual measured frequencies were 25.205, 21.067, 18.131, 16.009 MHz respectively. The results show deviations of -0.226, -0.125, -0.189, -0.016 MHz respectively between the calculated and measured values. It is anticipated that the actual frequency is slightly lower than the calculated one due to the fact that Wheeler's equation only considers structural parasitic capacitance, but not the parasitic capacitance of the yarns. This additional capacitance is expected to lower the actual resonant frequency.

We then estimated the yarn parasitic capacitance for each coil by multiplying the yarn's total length and the extracted unit length parasitic capacitance of AMBERSTRAND® 166. This allowed us to compute an amended frequency using the equation $f = \frac{1}{2\pi\sqrt{L(C+C_p)}}$, where C_p is the calculated yarn parasitic capacitance. For instance, the coil with 6 turns has a length of 77.28 cm, generating a calculated parasitic capacitance of 0.273 pF ($77.28 \times 3.53 = 273 \text{ fF}$). The comparison of the amended frequency and measured frequency is presented in Table S6. In comparison to Table S5, the deviation has been reduced and the actual frequency is now much closer to the actual measured frequency, with an average deviation of only around 0.01 MHz.

Line 440-463 (revised manuscript):

“Inductance Calculation using Wheeler's Equation

The inductance of each coil was calculated using following equation [18]:

$$L_{circle} = 31.33\mu_0 N^2 \frac{a^2}{8a + 11c} = 31.33\mu_0 N^2 \frac{a}{8 + 22\rho} \quad (9)$$

In this equation, μ_0 represents the magnetic permeability of free space ($\mu_0 = 4\pi \times 10^{-7} \text{ Hm}^{-1}$), N is the number of turns in the coil, a is the average radius of the coil ($a = \frac{d_{out}+d_{in}}{4}$), c is the distance between the inner turn and outer turn, and ρ is defined as $\rho = \frac{d_{out}-d_{in}}{d_{out}+d_{in}}$.

Fabrication of NFC coils

The commercial software PE-DESIGN 10 was used to convert the digitalized stitch trajectory of the coil pattern. We fabricated several prototypes of the planar coils in the shape of circular using the digital embroidery process, as shown in Figure S3(a). The digital embroidery machine PR670E, along with the conductive yarns AMBERSTRAND® 166, which served as the bottom bobbin to support the upper textile thread, used in this process. The spiral coil prototypes were then embroidered on a flexible cotton substrate, as shown in Figure S3(b).

Measurement of NFC Resonant Frequency

After the e-textile coils were fabricated using conductive yarns, we investigated their radio frequency (RF) properties. By connecting a capacitor to the terminals of the embroidered coils, we were able to create an NFC resonant tag. We used the Wheeler's equations to estimate the inductance (L) of square and circular coil inductors with different geometrical parameters. With the calculated L of the inductors and the selected capacitance (C) values, we were able to determine the theoretical optimal operating frequency using the equation $f = \frac{1}{2\pi\sqrt{LC}}$, as listed in Table S5-8.

To measure the self-resonance frequency of the embroidered coils with capacitors, we used the HF/LF Tag Test machine (QBG5C, AI YI). We connected capacitors to the terminals of several prototypes to account for individual variations. Once the tag devices were placed steadily on the test platform, we were able to read the resonant frequency value directly from the platform, As shown in Figure S3(c).”

Line 565-569 (revised manuscript):

“[28] Uddin, M. M. et al. Highly flexible and conductive stainless-steel thread based piezoelectric coaxial yarn nanogenerators via solution coating and touch-spun nanofibers coating methods. *Smart Materials and Structures* 31, 035028 (2022).

[29] Jiang, F. et al. Stretchable, breathable, and stable lead-free perovskite/polymer nanofiber composite for hybrid Triboelectric and piezoelectric energy harvesting. *Advanced Materials* 34, 2200042 (2022).”

4. There is one major difference between the simulation results shown in Figure 3(c) and the measurement results shown in Figure 4(d). In the simulation, the waves become larger as the frequency increases, while they become smaller in the measurement results. I cannot determine whether this is due to the influence of the real environment or to incomplete mathematical modeling. I would appreciate an appropriate discussion.

Reply: Thank you for this question. We apologize for not clearly describing the purpose of Figure 3(c) in the original manuscript. The purpose of this figure is to demonstrate that the influence of C_{pti} on the results is minimal and can be safely ignored in the final turn-to-turn model. The curve shown in Figure 3(c) is not optimized to fit the measurement data because we intentionally held all parameters constant except for the presence of C_{pti} . This is why it appears differently from Figure 4(d).

It is also worth noting that the amplitude of the impedance curve is influenced by the resistance of the device, which determines the Q-factor. However, this parameter does not impact the desired results of our experiment. In fact, the resistance used in Figure 3(c) is different from the resistance in the data shown in Figure 4(d). In Figure 4(d), we are presenting the impedance curve of 0.32 mm copper-wire-based DUTs for different numbers of turns.

We apologize for any confusion caused by our lack of clarity in the original manuscript. We have revised the text to better explain the purpose and significance of these figures in the revised manuscript.

Line 184-185 (revised manuscript):

“All parameters are held constant except for C_{pti} .”

Line 232-236 (revised manuscript):

“The trend of the change in amplitude of the curves is irrelevant to the study because the amplitude is influenced by the resistance of the device, which determines the Q-factor and this factor does not impact the desired results of the experiment. Using the results from Table 2, the lump-sum capacitances can be calculated from Equation (1).”

5. It should be clearly stated how both ends of the yarn were treated. The statement is necessary to ensure reproducibility.

Reply: Thank you for this valuable suggestion. In our experiments, we connected both ends of the yarn to the port extension. To maintain consistency in our measurements, we set the distance between the ends of the helical inductors and the extension to be equal to the length of half a turn of the helical inductor. This ensures that the extension conditions are the same for all measurements, which helps to preserve reproducibility. We apologize for not clearly describing this step in our original manuscript. We have now added this information to the Methods section in the revised manuscript. Thank you again for your suggestion and for helping us to improve the clarity of our work.

Line 426-432 (revised manuscript):

“The extension used in our experiments consists of a jumper wire, two wire clamps, and an SMA connector. One end of the jumper wire is soldered onto the SMA connector, while the other end is connected to one of the wire clamps. To maintain consistency in our data, we set the distance between the ends of the helical inductors and the extension to be equal to the length of half a turn. Additionally, we included the extension in the calibration process (using the Port Extension mode on the VNA) to minimize its impact on the measurements.”

6. The scales of the three pictures in Figure 4(a) are different and should be the same. If there is no special reason, the backgrounds should also be the same. It is difficult to tell the difference when they are placed side by side.

Reply: Thank you for this suggestion. You are correct that using the same scale will clearly illustrate the differences between the yarns. We have made the necessary changes to the figures and included them in the revised manuscript at the same scale.

Line 243 (revised manuscript):

7. There is an unnecessary closing bracket on line 131.

Reply: Thank you for pointing out this mistake. We have corrected this error and removed the bracket as requested.

Line 145-146 (revised manuscript):

“Here only the capacitance between adjacent turns, i.e., are i.e., $C_{t(i,i+1)}$ and $C_{t(i+1,i+2)}$ are considered, and the capacitance between turns of larger separation can be ignored.”

Reviewer #2 (Remarks to the Author):

The paper investigates a fundamental issue of parasitic capacitance which exists by nature in most applications with conductive materials. In particular, the paper is focused on the discovery of parasitic capacitance in electronic textiles sector where conductive yarns are commonly used. The paper presents two models, lump-sum and turn-to-turn models, to extract the parasitic capacitance with aid of a MATLAB modelling and experiences. Three commercially off the shelf conductive yarns have been tested in addition to their countermodels made using copper wires. Overall capacitance and unit capacitance have been derived from these methods and authors claims the models can be used to estimate parasitic capacitance in other scenario.

Reply: Thank you for your review of our manuscript. We appreciate your summary of the main focus and contributions of the paper. As you mentioned, our paper investigates the fundamental issue of parasitic capacitance in conductive materials, with a particular focus on the electronic textiles sector where conductive yarns are commonly used. We present two models, the lump-sum model and the turn-to-turn model, which use MATLAB modeling and experimental data to extract the parasitic capacitance of conductive yarns. We have tested these models using three commercially available conductive yarns, as well as countermodels made with copper wire. From these experiments, we have derived the overall capacitance and unit capacitance of the yarns and suggest that these models can be used to estimate parasitic capacitance in other scenarios.

We appreciate your thorough review of the manuscript and hope that our work will be considered for publication. We have included the revised content in this submission and have highlighted it in the revised manuscript for your convenience. Thank you again for your valuable feedback.

However, the reviewer has the following concerns regarding this manuscript:

1. The paper claims that “parasitic capacitance greatly affects the performance of e-textile device in RF and high frequency applications” but fails to explore in such frequency bands. The maximum frequency explored in the paper is 100MHz which is well below common RF bands. Since the capacitance is frequency dependent, the paper should investigate the impact of parasitic capacitance in the correct RF bands.

Reply: Thank you for your question about the relevance of our study to RF and high frequency applications. It should be noted that the application frequency may differ from the test frequency used in our experiment. Actually, the parasitic capacitance of the conductive yarns is an intrinsic property of the material that is influenced by factors such as permittivity, yarn diameter, and spacing, rather than frequency. Therefore, it is not affected by the frequency of the application. In our experiment, we chose a testing frequency range of 100kHz to 100MHz because the resonant frequency of the system (created by the combination of the helical inductor and the conductive yarn parasitic capacitor) falls within this range. We use the resonant frequency to extract the parasitic capacitance, which is therefore independent of the application frequency.

RF and high frequency applications such as NFC coils (Near Field Communication, 13.56 MHz), RFID antennas (900 MHz and 2.45 GHz), and transmission lines all operate within the RF band. In the revised manuscript, we will include an additional case study of the potential application to

provide more details on how the parasitic capacitance may impact these applications and why our research is relevant in these contexts. Thank you again for raising this valuable question.

Line 221-225 (revised manuscript):

“A testing frequency range of 100kHz to 100MHz is chosen because the resonant frequency of the system (created by the combination of the helical inductor and the conductive yarn parasitic capacitor) falls within this range. It should be noticed that the application frequency may differ from the test frequency used in our experiment.”

2. Following the Q1, how significant the parasitic capacitance is in e-textile application and what impact the parasitic capacitance will be? The paper should compare and analyse in a case study to demonstrate the importance of parasitic capacitance in e-textile sector. The significance needs to be quantified so the method can greatly benefit a wider community.

Reply: Thank you for your suggestion to include a case study on how the parasitic capacitance may impact in e-textile sector. In the revised manuscript, we have selected the topic of designing NFC coils using conductive yarn as a case study to illustrate the impact of parasitic capacitance on RF applications.

In the design of NFC coils, traditional methods such as Wheeler's equation typically assumes the use of a metallic conductor. However, when using conductive yarns to construct NFC coils, the presence of parasitic capacitance may result in deviations from the desired inductance value. To study this effect, we have used Wheeler's equation to design several planar coils that do not consider the parasitic capacitance of the conductive yarn. We will include this case study in the revised manuscript to provide more information on the impact of parasitic capacitance on RF applications and to demonstrate the relevance of our research to these types of applications.

Line 309-342 (revised manuscript):

“Designing NFC Coils with Conductive Yarn

As a case study that incorporates the estimated parasitic capacitances of conductive yarns, we will calibrate the resonant frequency of embroidered NFC coils that are commonly used in smart apparel applications. Yarn-based NFC devices are getting increasingly popular in wearable electronics in recent years as they benefit from the flexibility and breathability [28,29] of the fabric-based structures. Traditional inductance calculation method for NFC coils uses Wheeler's equation, which is based on the assumption of metallic conductors. As we have demonstrated previously, the use of conductive yarns may lead to deviations in the desired inductance induced by the existence of additional parasitic capacitances.

To evaluate this effect, several planar spiral coils with varying number of turns (5, 6, 7, 8) were designed and fabricated using AMBERSTRAND® 166 conductive yarns. The detailed design parameters can be found in the supplementary materials (Figure S2 and Table S4). The inductance of each coil was calculated using Wheeler's equation.

To emulate the functionality of an NFC device, we created an NFC resonant tag by connecting a capacitor to the embroidered coils. The calculated inductance (see Methods Section) of the coils and the capacitance of the external capacitor were used to calculate the theoretical optimal operating frequency, which is given by the equation $f = \frac{1}{2\pi\sqrt{LC}}$. We then measured the actual resonant frequency of the tag using an NFC testing platform.

We then estimated the yarn parasitic capacitance for each coil by multiplying the yarn's total length and the extracted unit length parasitic capacitance of AMBERSTRAND® 166. This allowed us to compute an amended frequency using the equation $f = \frac{1}{2\pi\sqrt{L(C+C_p)}}$, where C_p is the calculated yarn parasitic capacitance. For instance, the coil with 6 turns has a length of 77.28 cm, generating a calculated parasitic capacitance of 0.273 pF ($77.28 \times 3.53 = 273 \text{ fF}$). The comparison of the amended frequency and measured frequency is presented in Table S6. In comparison to Table S5, the deviation has been reduced and the actual frequency is now much closer to the actual measured frequency, with an average deviation of only around 0.01 MHz."

Line 440-463 (revised manuscript):

"Inductance Calculation using Wheeler's Equation

The inductance of each coil was calculated using following equation [18]:

$$L_{circle} = 31.33\mu_0 N^2 \frac{a^2}{8a + 11c} = 31.33\mu_0 N^2 \frac{a}{8 + 22\rho} \quad (9)$$

In this equation, μ_0 represents the magnetic permeability of free space ($\mu_0 = 4\pi \times 10^{-7} \text{ Hm}^{-1}$), N is the number of turns in the coil, a is the average radius of the coil ($a = \frac{d_{out} + d_{in}}{4}$), c is the distance between the inner turn and outer turn, and ρ is defined as $\rho = \frac{d_{out} - d_{in}}{d_{out} + d_{in}}$.

Fabrication of NFC coils

The commercial software PE-DESIGN 10 was used to convert the digitalized stitch trajectory of the coil pattern. We fabricated several prototypes of the planar coils in the shape of circular using the digital embroidery process, as shown in Figure S3(a). The digital embroidery machine PR670E, along with the conductive yarns AMBERSTRAND® 166, which served as the bottom bobbin to support the upper textile thread, used in this process. The spiral coil prototypes were then embroidered on a flexible cotton substrate, as shown in Figure S3(b).

Measurement of NFC Resonant Frequency

After the e-textile coils were fabricated using conductive yarns, we investigated their radio frequency (RF) properties. By connecting a capacitor to the terminals of the embroidered coils, we were able to create an NFC resonant tag. We used the Wheeler's equations to estimate the inductance (L) of square and circular coil inductors with different geometrical parameters. With the calculated L of the inductors and the selected capacitance (C) values, we were able to determine the theoretical optimal operating frequency using the equation $f = \frac{1}{2\pi\sqrt{LC}}$, as listed in Table S5-8.

To measure the self-resonance frequency of the embroidered coils with capacitors, we used the HF/LF Tag Test machine (QBG5C, AI YI). We connected capacitors to the terminals of several prototypes to account for individual variations. Once the tag devices were placed steadily on the test platform, we were able to read the resonant frequency value directly from the platform, As shown in Figure S3(c).”

Line 565-569 (revised manuscript):

[28] Uddin, M. M. et al. Highly flexible and conductive stainless-steel thread based piezoelectric coaxial yarn nanogenerators via solution coating and touch-spun nanofibers coating methods. *Smart Materials and Structures* 31, 035028 (2022).

[29] Jiang, F. et al. Stretchable, breathable, and stable lead-free perovskite/polymer nanofiber composite for hybrid Triboelectric and piezoelectric energy harvesting. *Advanced Materials* 34, 2200042 (2022).

3. The model of parasitic capacitance in a conductive yarn, shown in Fig 1 (c), is questionable. Depending on the composite of yarns, there should not be any capacitance between two conductive filaments if it is made of stainless steel. Under twisting process of filament during the yarn manufacturing pushes conductive filaments together. This will at a result, create a conductive path between two filaments, assuming the filament is 100% stainless steel. This conductive path will therefore eliminate any capacitor being formed. The paper seems to assume that there is an air gap between two filaments so capacitance exists. This is not a case due to the reason explained above. Authors needs to explain more about materials with the support of any assumptions.

Reply: Thank you for bringing this issue to our attention. We appreciate your insight and agree that the twisting of conductive filaments is an important consideration in our model. You are correct that these filaments will form a conductive path when twisted together, which can result in a short circuit and the elimination of the capacitor between them. However, as we observed in the SEM image of the conductive yarn (Figure 1(a)), the yarn is only sparsely twisted and there are still gaps between the filaments. Additionally, the filaments are not perfect conductors, and the resistance is proportional to their length. As the length of adjacent filaments between two contact points may vary, this can result in a differing distribution of voltage along the filaments, leading to the formation of parasitic capacitance. We apologize for not properly explaining this in the original manuscript. We will include a discussion on this issue in the revised manuscript.

Line 104-109 (revised manuscript):

“To be more specific, as illustrated in Fig. 1(a), when bundles of filaments are twisted, gaps and contacts are formed along the length of adjacent filaments. Filaments are resistive in nature and the length of adjacent filaments between contact points may not have the same length. The difference of resistance will lead to different distributions of voltage along the filaments, and will result in different potentials between adjacent filaments.”

4. Line 263, the paper claims that the length of each helical turn is the perimeter of a circle. This is only true when the space between two turns is much smaller compared to the diameter of the turning circle. However, this is not true with the setup of this paper. The 10 mm space between turns is compared to 44 mm of turn diameter. The unit turn length is not equal to the perimeter of 44 mm circle.

Reply: Thank you for your suggestion. We agree that treating every turn as an oval rather than a circle is a more reasonable approach. We have made the necessary changes to our calculations. The

equation for the perimeter of an ellipse is $L = 4 \int_0^a \sqrt{1 + \frac{b^2 x^2}{a^2(a^2 - x^2)}} dx$, and using this equation,

the updated length for each turn is 121.8 mm, or 12.18 cm. We have modified the length and the calculated unit parasitic capacitance accordingly.

Line 300-302 (revised manuscript):

“Where L_{turn} is the length of each turn, where $L_{turn} = 4 \int_0^a \sqrt{1 + \frac{b^2 x^2}{a^2(a^2 - x^2)}} dx = 12.18 \text{ cm}$ in our case. C_{up} for S310 is 2.71 fF/cm , while for S480 is 0.99 fF/cm , and for AMBERSTRAND 166, it has the largest C_{up} of 3.53 fF/cm .”

5. The measurement using VNA uses port extension with jumper wires which could introduce parasitic capacitance. How can this capacitance be minimised and then eliminated?

Reply: Thank you for your question. To ensure the accuracy of our measurements, we calibrated the VNA with a connecting extension before performing any measurements using the Port Extension mode. This helped to minimize any potential errors and ensure that our results were reliable. In addition, we calculated the parasitic capacitance of the conductive yarn by comparing it to the known capacitance of a copper wire. Both measurements were conducted under the same conditions, so any common-mode errors would be canceled out, ensuring that our results are accurate. We apologize for not including this information in the Methods section of the original manuscript and have now added these details to the revised version.

Line 426-432 (revised manuscript):

“The extension used in our experiments consists of a jumper wire, two wire clamps, and an SMA connector. One end of the jumper wire is soldered onto the SMA connector, while the other end is connected to one of the wire clamps. To maintain consistency in our data, we set the distance between the ends of the helical inductors and the extension to be equal to the length of half a turn. Additionally, we included the extension in the calibration process (using the Port Extension mode on the VNA) to minimize its impact on the measurements.”

6. Authors need to be careful when using vague wordings, such as “very insignificant”, “even more minimal” etc. Those wordings should not be in the paper unless being backed by the scientific evidence.

Reply: Thank you for your suggestion to improve the clarity of our manuscript. You are correct that the use of vague words is not scientific and we apologize for any confusion this may have caused. We have revised the manuscript to include more precise descriptions in place of these vague words.

Line 182-183 (revised manuscript):

“From simulation, we have found out that C_{pti} actually has a trivial contribution to the overall oscillating frequency of the helical inductor.”

Line 187-194 (revised manuscript):

“The results demonstrate that as the relative values of C_{pti} changes, the resonant frequency does not show any obvious changes. The resonant frequency only changes insignificantly. The contribution of C_{pti} becomes more minute as the number of turns increases. This phenomenon can be explained from the equivalent circuit model in the bottom of Figure 3(c), as the number of turns increases, capacitances C_{pti} of different turns are connected in series of each other, more turns will cause more C_{pti} to be series-connected, and their overall contribution will be minimized. In fact, when the turn number is over 10, the contribution of C_{pti} can be ignored.”

Line 237-240 (revised manuscript):

“It also shows the copper wire gauge had little effect on the lump-sum structural capacitances, as the results from 0.32 mm copper wire are similar (within measurement margin) from those of 0.50 mm copper wire.”

7. Some other points of considers:

- a. Line 20: what is meant by “yarn`s microstructure”?
- b. Line 53: “high frequency RF applications” have not been defined.
- c. Line 79: why S310, S480 and 166 are used and how can these three commercial yarns represent the parasitic capacitance of overall conductive yarns? Have you considered filament dimension in connection with the yarn dimension?

- d. Line 114: stop using “etc” when describing the notation and meaning of symbol. Just say what are they.
- e. Line 132: “capacitance between two turns of large separation can be ignored”. Please define how large is large and what is the assumption here.
- f. Line 160: figure 3(c) needs to be improved in terms of its presentation. Colouring is not a good option when viewing in black/white. Suggesting to add symbols to each curve.
- g. Line 269: typo. Should it be “comparison” not “caparison”?
- h. Line 306: How lump-sum resistance is measured using a four-point-probe?

Reply: Thank you for bringing these questions to our attention. Below, we have provided individual responses to each of the points you raised. And necessary revisions are made in the revised manuscript.

- a. The yarn's microstructure refers to the structure of the yarn when viewed under a microscope. It includes the individual fibers or filaments that make up the yarn, as well as any other elements that may be present, such as coatings or twist. The microstructure of the yarn can have a significant impact on the physical properties of the yarn, such as its strength, elasticity, and conductivity. Examining the yarn's microstructure can be useful in understanding how the parasitic capacitance is formed inside the yarn.
- b. "High frequency RF applications" refer to applications that operate within the radio frequency (RF) spectrum, typically at frequencies above 1 MHz. RF applications may include NFC (Near Field Communication), RFID (Radio Frequency Identification), and embroidered conductive-yarn-based interconnects that transmit high-frequency signals in smart apparel applications.
- c. Conductive yarns can be classified into three categories based on the types of fibers they are composed of: pure electrically conductive metallic fibers (such as stainless steel), intrinsic conductive polymer fibers, and conductive polymer composite fibers. Our selection of S310, S480, and 166 covers these categories, providing a representative range of options. S310 and S480 are both made of stainless steel and have the same filament dimensions, but they differ in the number of fibers twisted together.
- d. Thank you for this suggestion. We apologize for using "etc" when describing the notation and meaning of symbols. Instead, we have listed all of the symbols and their meanings in the revised manuscript for clarity.
- e. We apologize for any confusion regarding this concept. The distance between two adjacent turns in our model is 10 mm, which is much greater than the diameter of the yarn. According to the equation for capacitance, the capacitance is inversely proportional to the separation distance. In our turn-to-turn model, we are examining the capacitance between two adjacent turns. We assume that when the distance between the turns is greater than 10 times the diameter of the yarn, it is large enough to be ignored in our calculations.
- f. Thank you for suggesting the addition of symbols to Figure 3(c). In the original paper, we used colors and different types of lines (dash, solid, and dash-point) to distinguish the curves, as they are already quite close to each other. We will do our best to make the figure more presentable.

- g. You are correct, the word should be "comparison." We will make the necessary correction. Thank you for your help in improving the quality of our work.
- h. To measure the lump-sum resistance, we applied a DC current (I) between the outer two probes and then measured the voltage drop (ΔV) between the inner two probes. We calibrated the measurement to account for contact resistance. The resistance was calculated by dividing the voltage drop (ΔV) by the current (I).

Line 50-54 (revised manuscript):

“More recent research began to investigate the unique design challenges of e-textile devices [24-26]. However, these works mostly focus on the resistive properties of the conductive yarns, the reactive properties, which determine the devices’ performance in high frequency and RF applications (typically at frequencies above 1 MHz), had not been systematically investigated.”

Line 84-86 (revised manuscript):

“We selected these conductive yarns because they represent a wide range of categories within the conductive yarn market.”

Line 127-129 (revised manuscript):

“The helical structure can be defined by the number of turns N , diameter of the core r , the separation between the adjacent turns p , and the width of grooves δ .”

Line 147-148 (revised manuscript):

“We assume that when the distance between the turns is greater than 10 times the diameter of the yarn, it is large enough to be ignored in our calculations.”

Line 170-172 (revised manuscript):

“Each turn in the helical structure is identical to each other, therefore, the value of these parameters is the same between turns, for example ($R_1 = R_2 = \dots = R_i$).”

Line 306-308 (revised manuscript):

“Nevertheless, the relative comparison of the parasitic capacitances between these three yarns are consistent between the lump-sum model and the turn-to-turn model.”

Line 378-381 (revised manuscript):

“For example, to measure the lump-sum resistance, we applied a DC current (I) between the outer two probes and then measured the voltage drop (ΔV) between the inner two probes. We calibrated the measurement to account for contact resistance. The resistance was calculated by dividing the voltage drop (ΔV) by the current (I).”

Line 177 (revised manuscript):

“

”

Reviewer #3 (Remarks to the Author):

Very interesting topic! I myself have been thinking about the reactive properties of electrically conductive yarns. I also applaud the will to characterise these kinds of yarns in a systematic way without having a specific application in mind. Having said that, I would argue that most of the time these conductive yarns are used to form either woven, knitted or embroidered structures where many conductive yarns are interconnected. In such situations, I believe that the dominating capacitances would either be those between sets of yarns in close proximity to each other or between a set of yarns and other bodies (e.g. ground). I did very rudimentary measurements (frequency response analysis) of conductive fabrics and yarns to see if the mechanical tensile load would affect the impedance of the yarn or fabric. I swept the frequency between 10 Hz up to 10 MHz and applied different tensile loads on the samples and I could not see any change apart from the resistance of the samples. I think that as long as one does not have any significant inductance in the samples, the potential variation of the stray or structural capacitances in yarns or simple fabrics themselves have a very minute effect on the impedance.

Reply: We are grateful for your interest in our topic and your efforts to reproduce our results. We sincerely apologize for the confusion and difficulty you experienced while attempting to reproduce our work. Unfortunately, we made a mistake in the original paper - we used incorrect labels in Table 4, which led you to input the wrong parameters for your simulation (further explanations are provided below).

The parasitic capacitance of conductive yarns is caused by the gaps between the individual filaments in a single yarn. While it is not as significant as the capacitances between yarns themselves or between yarns and bodies, it does contribute slightly to the proximity capacitance. Thanks for noticing this. It is true that when sweeping the frequency between 10 Hz and 10 MHz, there may not be a noticeable difference between the impedance plots of conductive yarns and perfect electric conductors. Our experiments have also shown this. However, in our measurements, the frequency range was set from 10 Hz to 100 MHz in order to extract the resonant frequency of the DUTs. The difference in resonant frequency between conductive yarns and Perfect Electric Conductor (PEC)s demonstrates the presence of parasitic capacitance in the conductive yarns.

We are grateful for your comprehensive review of the manuscript. We have incorporated your suggested revisions in this submission and highlighted them in the revised manuscript for your convenience. Thank you again for your helpful feedback.

1. I do not wholly understand your equivalent circuit in Figure 1 d though. Looking at Figure 1 a it seems to me that the filaments touch each other at the ends of the gaps indicated. That, to me, would suggest that there should be a contact resistance in parallel with the parasitic capacitances (see my annotated version of your Figure 1). Unless of course that each filament has an insulating layer on its surface. On line 100, you state: "Because of the intrinsic resistance of the yarn materials, adjacent filaments may not have the same potential." I could not find a reference to any data sheet about the yarns you used, it could be good if you added that on lines 81-83.

Reply: Thank you for bringing this issue to our attention. Firstly, we want to clarify the purpose of the circuit model shown in Figure 1(d). Unlike the turn-to-turn model, this circuit model in Figure

1(d) is an equivalent circuit model for a section of the conductive yarn. Its purpose is to help the reader understand why parasitic capacitance exists on a piece of conductive yarn. On the other hand, the turn-to-turn model shown in Figure 3 and the lump-sum model in Figure 2 are models based on the helical inductor made from the conductive yarn. Therefore, we cannot use the model in Figure 1 to understand the lump-sum and turn-to-turn models. Additionally, the lump-sum model and the turn-to-turn model also have different purposes. The lump-sum model is used to approximate the total parasitic capacitance of the helical conductive yarn, while the turn-to-turn model is used to accurately determine the parasitic capacitance for each turn and can be used to calculate the parasitic capacitance per unit length. In the revised manuscript, we have added the contact resistance in Figure 1(d). This does not impact the conclusion of the paper.

Regarding the statement on line 100, the filaments in the conductive yarn are not perfect conductors and their resistance is proportional to their length. As the length of adjacent filaments between two contact points may vary, this will result in a differing distribution of voltage along the filaments. This potential difference, combined with the separation gap between the adjacent filaments, leads to the creation of parasitic capacitance. For example, as depicted in the graph below, assume R_1 , R_2 belong to the upper filament, R_3 , R_4 belong to the lower filament, and C_1 is the equivalent capacitance in between. The length of the filaments is different, as mentioned earlier, so R_1 , R_2 , R_3 , R_4 have different values. If we assume that the upper filament is connected to a voltage and the lower filament is connected to the reference ground, it is clear that there will be a voltage difference on either side of C_1 . This results in the creation of parasitic capacitance for C_1 . Thus, the parasitic capacitance of the conductive yarn is caused by different resistance values due to length differences between the contact points of adjacent filaments.

Line 104-109 (revised manuscript):

“To be more specific, as illustrated in Fig. 1(a), when bundles of filaments are twisted, gaps and contacts are formed along the length of adjacent filaments. Filaments are resistive in nature and the length of adjacent filaments between contact points may not have the same length. The difference of resistance will lead to different distributions of voltage along the filaments, and will result in different potentials between adjacent filaments.”

Line 111-120 (revised manuscript):

“A distributed equivalent circuit model of the parasitic capacitances created by the gaps is illustrated in Figure 1(c), where R_{seg1_top} and R_{seg2_top} , etc. are the resistance of the filament segments that form the top plate of parasitic capacitances between two contact points, and R_{seg1_bottom} , R_{seg2_bottom} , etc., are the resistance of the filament segments that form the bottom plate. R_c is the contact resistance formed by the adjacent filaments touching each other at the ends of the gaps. C_{gap1} and C_{gap2} are the parasitic capacitances created by the gaps.

The capacitance formed by each tiny gap can be consolidated into parasitic capacitance between adjacent filaments. Figure 1(d) illustrates a cross section of the conductive yarn and the collective parasitic capacitances between the filaments.”

Line 81-84 (revised manuscript):

“We have measured the parasitic capacitance of three different conductive yarns, namely, S310 (stainless steel yarn with diameter of 0.31 mm), S480 (stainless steel yarn with diameter of 0.48 mm) and AMBERSTRAND® 166 [32] (conductive polymer composite yarn with diameter of 0.25 mm), the unit-length capacitance (in term of fF/cm) are 2.71, 0.99 and 3.53.”

Line 574-576 (revised manuscript):

“[32] AMBERSTRAND 166 Datasheet Available at: <https://static1.squarespace.com/static/558431b9e4b0875de16c5494/t/5d9d011bb7bacf1e9e34d93c/1570570527783/Amberstrand+166.pdf>.”

Line 111-117 (revised manuscript):

“A distributed equivalent circuit model of the parasitic capacitances created by the gaps is illustrated in Figure 1(c), where R_{seg1_top} and R_{seg2_top} , etc. are the resistance of the filament segments that form the top plate of parasitic capacitances between two contact points, and R_{seg1_bottom} , R_{seg2_bottom} , etc., are the resistance of the filament segments that form the bottom plate. R_c is the contact resistance formed by the adjacent filaments touching each other at the ends of the gaps. C_{gap1} and C_{gap2} are the parasitic capacitances created by the gaps.”

Line 92 (revised manuscript):

2. So are the filaments in contact with each other along the yarn or are they only in contact at the terminals? If they are in contact along the yarn then I would like an explanation of how you neglect the contact resistance between the filaments. If they are not then I would believe that the part C_{pti} (on line 165) would make a large contribution to the overall capacitance. In addition, if the filaments are in contact with each other then how uneven is the conductivity of the medium? Because if the length of the gaps (not the distance between the conductive surfaces) is of the order of $100\ \mu\text{m}$, then how likely is it that two filaments have different potential (i.e. is the relaxation time of the conductive medium greater than, say $1/1e^8 = 10\ \text{ns}$)? For a homogeneous piece of copper, I think that the relaxation time is of the order of $1e-18\ \text{s}$.

Reply: Thank you for your question. We consider the filaments to be in contact with each other along the yarn, but this does not affect our theory on the existence of parasitic capacitance, as explained above. In our previous discussion, the parasitic capacitance exists due to the inequality in filament lengths between contact points. The contact resistance does not significantly contribute to the parasitic capacitance. The circuit in Figure 1(d) shows that the parasitic capacitance is evenly distributed along the conductive yarn. However, it is important to note that this is not the circuit model we are using in the later parts of the paper. Rather, it simply illustrates that the total parasitic capacitance increases as the conductive yarn becomes longer. In the revised manuscript, we reemphasized the origin of the parasitic capacitance of conductive yarn.

Line 104-109 (revised manuscript):

“To be more specific, as illustrated in Fig. 1(a), when bundles of filaments are twisted, gaps and contacts are formed along the length of adjacent filaments. Filaments are resistive in nature and the length of adjacent filaments between contact points may not have the same length. The difference of resistance will lead to different distributions of voltage along the filaments, and will result in different potentials between adjacent filaments.”

3. Again about possible contacts between the filaments: the reasoning on lines 145 – 149 leading to the statement that the micro-structural capacitances could be viewed as connected in parallel to the turn-to-turn capacitance of the coil. If such contacts exist then perhaps the yarn could be modelled as a parallel R-C link per unit length, and if so, should not C_{pti} be in parallel with R_i rather than with the series connection of R_i and L_i (as in Figure 3b)?

Reply: Thank you for your question. In our study, we examined the impact of C_{pti} when connected in parallel with R_i , rather than in series with R_i and L_i . Figures A and B show the corresponding MATLAB models for these configurations, and Figures C and D present the simulation results for 8 turns using the data in Table 4 and $C_{pti} = C_{pgi}$. The 'old' labeled curves correspond to the model in Figure B, while the 'new' labeled curves correspond to the model in Figure A. Figure D is a magnified view of the peak of the real part in Figure C as highlighted in red. We care about this part because it reveals the difference of the resonant frequencies. As Figure D shows, there is little difference between these curves, indicating that the position of C_{pti} has a minimal effect on our results. Therefore, we can conclude that the position of C_{pti} is only a minor consideration in our study.

Figure A

Figure B

Figure C

Figure D

Line 194-195 (revised manuscript):

“This conclusion remains valid when C_{pti} is connected in parallel with R_i .”

4. On line 156 you state that the C_{pi} may contribute to C_{ti} and C_{gi} , how does it contribute to C_{tgi} ?

Reply: Thanks for your question. We apologize if we have misunderstood your question, but we do not have C_{tgi} in our analysis. If you are referring to C_{pgi} and C_{pti} , we do believe the parasitic capacitance would contribute to both C_{ti} and C_{gi} . As depicted in Figure 3(a), the microstructure-induced capacitances (C_1, \dots, C_n) are formed by the filament-to-filament plating structure where the filaments are topologically parallel to each other. The filament-induced capacitance can be considered to be connected in parallel with the lump-sum capacitance induced by the helical structure. The total conductive yarn-induced parasitic capacitance C_p is connected in parallel with the structural capacitance C . The parasitic capacitance generated by the microstructure of the yarn, C_{pi} , can contribute to both the turn-to-turn capacitance C_{ti} and the turn-to-ground capacitance C_{gi} . Therefore, we denote the microstructure-based capacitance as C_{pgi} and C_{pti} and connect them in parallel with the structural turn-to-turn capacitance C_{ti} and turn-to-ground capacitance C_{gi} , respectively, as shown in Figure 3(b). In the revised manuscript, we have made necessary revisions on this matter.

Line 172-176 (revised manuscript):

“The parasitic capacitance C_{pi} that is produced by the microstructure of the yarn can affect both the turn-to-turn capacitance C_{ti} and the turn-to-ground capacitance C_{gi} . To account for this, we can consider the microstructure-based capacitances C_{pti} and C_{pgi} in parallel with the structural turn-to-turn capacitance C_{ti} and turn-to-ground capacitance C_{gi} , respectively, as illustrated in Figure 3(b).”

5. About Figure 3d, 4d and 6, are the graphs absolute value and phase angle of the impedance or is it the real and imaginary parts of the impedance? In either case, it could be nice to write that out in the text somewhere.

Reply: We apologize for any confusion. In these figures, both the real and imaginary parts of the impedance share the same y-axis. Descriptions have been added to clarify this.

Line 186-187 (revised manuscript):

“The results of the real and imaginary part of the impedance are illustrated in Figure 3(c).”

Line 231-232 (revised manuscript):

“The curves represent real and imaginary part of the impedance. The hyperbola shaped curve is the imaginary part and the other one is the real part ”

Line 245-246 (revised manuscript):

“(d) Real and imaginary part of the impedance of the copper wire (0.32 mm) based DUTs with different number of turns.”

Line 291-292 (revised manuscript):

“Measurement (solid lines) V.S. Simulation (dash lines) results of the real and imaginary part of the impedance for (a) copper wire (0.32 mm) based DUTs, (b) S310 based DUTs, (c) S480 based DUTs, and (d) AMBERSTRAND® 166 based DUTs.”

6. So if I understand correctly the C_{ti} does not contribute in any significant way to the overall impedance. That is also what I found when I did some SPICE simulations using first your model (Figure 3b) and the values you report (and later on your simplified model in Figure 3d). I am a little bit confused about what the different reported values represent though. Using the values of Table 2 for 8 turns of the yarn S310 straight off and in addition the value for C_{S310}^p for 8 turns in Table 3 as 0.0199pF, one would get the following situation with your simplified tur-to-tur model (Figure 3d):

And running the simulation one gets the following spectra:

I did vary the C_{pg} between 20-50 fF. Neither of those peaks resemble the ones seen in Figure 3c, 4d or 6b. So I guess I didn't interpret your text correctly.

Reply: We apologize for any confusion caused by the incorrect labels in our manuscript. Both Table 2 and Table 3 are relevant to the lump-sum equivalent circuit of the helical air-core inductor depicted in Figure 2(b). Table 4 contains the values for the parameters of the simplified turn-to-turn circuit model for helical inductors for each turn. The correct values for the test case in question should be $C_{ti} = 7.5$ pF and $C_{gi} = 0.333$ pF, as indicated in Table 4. We have revised the manuscript to clarify this point. We also conducted simulations using SPICE to verify our results. The simulation results for 8 turns of yarn S310, using the values in Table 4, were consistent with the results obtained using MATLAB. This demonstrates that the functionality of MATLAB is equivalent to that of SPICE simulations, and that the results are reproducible. We apologize for any confusion caused by our misleading table. It is worth noting that C_{ti} does not significantly impact the overall impedance.

Line 295 (revised manuscript):

“Table 4. Parameters for each turn of simplified turn-to-turn circuit model of helical inductors (Figure 3(d)).

R_i^* (Ω)	L_i (μH)	C_{ti} (pF)	C_{gi} (pF)	C_{pgi} (pF)
”				

7. On lines 182-184 you state that the values for each R_i and L_i in the model can be taken as the measured total R and L divided by the number of turns. So using the yarn labelled S310 with 8 turns one would, from table 2, get $R_i = 3.36 \Omega/\text{turn}$ and $L_i = 0.256 \mu\text{H}/\text{turn}$. And using Equations 4-6 I get $C_{copper} = 2.1430$ pF and $C_{yarn} = 2.1450$ pF giving $C_p = 2$ fF. Now this C_{yarn} and C_p I take it is for the entire circuit so it should also be divided by the number of turns, right? In that case one $C_{yarn,i} = 0.268$ pF/turn and $C_{pi} = 0.25$ fF/turn. Then one gets the following:

And the impedance spectra as:

Reply: We apologize for any confusion caused by the incorrect use of Equations 4-6. These equations are only applicable to the lump-sum model and cannot be used for the turn-to-turn model. The values in Table 2 are related to the lump-sum model and cannot be applied to the turn-to-turn model. The turn-to-turn model is much more complex, as the structural capacitances are distributed and interconnected with other turn-to-turn elements such as L_i and R_i , and both the turn-to-turn capacitance and turn-to-ground capacitance are unknown. These variables cannot be determined simply by measuring the resonant frequencies. The lump-sum model and the turn-to-turn model are significantly different, as the lump-sum model is used to study the overall parasitic capacitance of the helical inductor, while the turn-to-turn model is used to study the parasitic capacitance on the unit length of each turn in the helical inductor.

To determine the variables in the turn-to-turn model, we propose a multi-variable nonlinear regression method based on impedance frequency response measurements at different numbers of turns. This extraction method involves two stages, as illustrated in Figure 5. When estimating the parameters, only L_i has a clear relationship with the lump-sum model and can be estimated using Equation (2) by taking the measured total L and dividing it by the number of turns. All of the other parameters are determined using a Genetic Algorithm. It is worth noting that, as mentioned in the manuscript, the resistance of each turn varies with the number of turns (which affects the Q factor), so we use the average value in the parameter estimation. We also conducted simulations using SPICE to verify the results, as mentioned in a previous response. The simulation results are shown in the graph above. We apologize for any confusion caused by our misleading table and have added additional description to improve the clarity of the differences between the two models.

Line 262-272 (revised manuscript):

“To summarize, the lump-sum model is used to analyze the overall parasitic capacitance of the helical inductor device, while the turn-to-turn model is used to study the parasitic capacitance of the conductive yarn on a unit length within the helical inductor. Lump-sum model is not a simple summation of the distributed model, as the lump-sum model does not take into account the distributed parasitic capacitance inside the helical coil structure. The only element that are commonly used by the two models is the inductance, where the turn-to-turn inductance L_i can be calculated from the inductance of the lump-sum model using Equation (2). It is important to note that Equations 4-6 are only applicable to the lump-sum model and cannot be used for the turn-to-turn model. Similarly, the values in Table 2 pertain to the lump-sum model and cannot be applied to the turn-to-turn model.”

8. In essence, I have difficulties reproducing your graphs. So maybe you could describe in a more clearer way. The only way I could get an impedance graph that resembles the one in Figure 6 b was if I used the values from Table 2 and the C_{S310}^p for 8 turns in Table 3 but ten times smaller in a single lump-element model (i.e. 1.99fF instead of 19.9 fF). Like this:

Then I can get a peak that is situated at ca 75.8MHz, but the peak value of the real part is slightly more than two times the value of the peak in Figure 6b.

I am not familiar with the Genetic Algorithm that you use for extracting the turn-to-turn parasitic capacitances, so I do not have any input on that.

Reply: We apologize for any confusion caused by the misleading labels in the manuscript. These have been revised. The inability to reproduce the graphs may be due to these incorrect labels. The peak value of the real part in the simulation is slightly more than twice the value of the peak in Figure 6(b) because the lump-sum model is not used to fit the measurement curves. In the lump-sum model, we are only concerned with the resonant frequency.

9. On line 258 (Table 4) you introduce C_s and C_c , what are those? Also in Table 4 you say that the R_i for the copper wire is 2.81 Ω per turn, but in Table 2 you measured it to be between 0.0517 – 0.0663 Ω /turn (lowest for 18 turns and highest for 8 turns). How are the values in Table 2 and Table 4 related?

Reply: We apologize for the misleading labels of C_s and C_c . These have been corrected, as mentioned previously. Table 2 contains the parameters estimated using the lump-sum equivalent circuit shown in Figure 2(b). Table 4 contains the parameters estimated using the simplified turn-to-turn circuit model of helical inductors shown in Figure 3(d). The inductance of the two tables can be related using Equation (2) as explained in a previous response. However, the capacitance and resistance do not have a clear relationship between the two tables, as the parameters in Table 2 represent the total values for the conductive yarn helical inductor, while the parameters in Table 4 represent the values for an individual turn of the conductive yarn helical inductor.

10. On lines 83-86 you say. “It is also important to mention that, like many other parasitic parameter extraction techniques used for electronic devices. The extraction results are affected by many structural and ambient conditions and cannot be very accurate.” Then on lines 198-207 you describe how you got the values for the total resistance and inductance and also how you measured the resonance frequency. These measurements are then used, if I understand correctly, to extract the total equivalent capacitance using Equation (1). The errorbars in Figure 4e for the 0.32 mm copper wire at 8 turns seems to be around 50 fF top to bottom and in Table 3 the calculated value of C_{S310}^p for 8 turns is 19.9 fF. This, in addition to your statement on lines 267-269 that the extracted values for the turn-to- turn model do not add up to the lump-sum value makes me wonder if there might be some factor that is not taken into consideration in your model. Having said that, I think it is nice to see that, as you say, there I a correlation between the lump-sum extracted values and the turn-to-turn one.

Reply: It is important to note that the lump-sum model and the turn-to-turn model are significantly different. The lump-sum model is based on an RLC equivalent circuit, which means it can be used to determine the total parasitic capacitance of the conductive yarn on the helical inductor from the resonant frequency equations. In contrast, the turn-to-turn model takes into account the structural capacitances and their interconnection with other turn-to-turn elements such as L_i and R_i , and both the turn-to-turn capacitance and turn-to-ground capacitance are unknown. Essentially, the turn-to-turn model focuses on the parasitic capacitance of the conductive yarn on a unit length of a helical

inductor, while the lump-sum model considers the overall parasitic capacitance of the helical inductor device.

We have divided the models into the lump-sum and the turn-to-turn because the helical inductor structure and configuration are relevant to the distributed capacitance on the conductive yarn. However, the lump-sum model does not take into account the details of the helical inductor configuration. By only studying the lump-sum model, it is not possible to analyze the parasitic capacitance on the unit length of each turn in the helical inductor. Therefore, it is necessary to propose a more detailed turn-to-turn model. Additionally, the relationship between these two models is difficult to establish because they are based on different assumptions. We cannot assume that the turn-to-turn parasitic capacitance will simply add up to the lump-sum parasitic capacitance.

On top of that, as we mentioned, the absolute value of parasitic capacitance may not be able to be accurately measured, but the relative value may still be meaningful to us. In the revised manuscript, we include a case study where we use conductive yarn to construct planar spiral coils. Results of the case study illustrates that incorporating the parasitic capacitance of the yarn allows for a more accurate prediction of the operating frequency of the yarn-based NFC coil.

11. It would be good if you could specify the different graphs in Figure 6, just as you did in Figures 3c and 4d. I do also recommend you to use more differentiable colours or perhaps different line types. For all figures showing the impedance spectra, it would be good if you could utilise the right hand y-axis to state if it is phase angle or imaginary part of the impedance.

Reply: Thank you for your suggestion. We agree that it would be helpful to specify the different graphs in Figure 6 in the same way that we did for Figures 3(c) and 4(d). We will include this information in the revised manuscript.

We also appreciate your suggestion to use more distinct colors or line types to differentiate between the different graphs. We will consider this in the revised manuscript and will do our best to use clear and easily distinguishable visual elements to present the data.

As an additional note, for all of the impedance figures, the real and imaginary parts of the impedance share the same y-axis. We have added text to the context to clarify this. Thank you again for your helpful feedback and for helping us improve the clarity and effectiveness of our work.

Line 186-187 (revised manuscript):

“The results of the real and imaginary part of the impedance are illustrated in Figure 3(c).”

Line 231-232 (revised manuscript):

“The curves represent real and imaginary part of the impedance. The hyperbola shaped curve is the imaginary part and the other one is the real part ”

Line 245-246 (revised manuscript):

“(d) Real and imaginary part of the impedance of the copper wire (0.32 mm) based DUTs with different number of turns.”

Line 291-293 (revised manuscript):

“Figure 1. Measurement (solid lines) V.S. Simulation (dash lines) results of the real and imaginary part of the impedance for (a) copper wire (0.32 mm) based DUTs, (b) S310 based DUTs, (c) S480 based DUTs, and (d) AMBERSTRAND® 166 based DUTs.”

Line 243 (revised manuscript):

Line 290 (revised manuscript):

REVIEWER COMMENTS

Reviewer #1 (Remarks to the Author):

The authors' dedication to the revision of the proposed method has ensured that it is now fully trusted. I thank the authors for their efforts. I recommend publication of this paper.

Reviewer #2 (Remarks to the Author):

Authors have made amendments to the manuscript to address concerns raised by the reviewers. Much of extensive work have been done to provide evidence to the satisfactory level. However, some concerns have not been understood and explained properly and therefore the reviewer has listed them as follow:

1.This question has not been explained fully. “The paper claims that parasitic capacitance greatly affects the performance of e-textile device in RF and high frequency applications but fails to explore in such frequency bands. The maximum frequency explored in the paper is 100MHz which is well below common RF bands. Since the capacitance is frequency dependent, the paper should investigate the impact of parasitic capacitance in the correct RF bands.”

It is understood that the coli has been evaluated within a band ranging from 100kHz to 100MHz due to its system resonance. More specifically, the coli has been evaluated from 30MHz to 80MHz. Authors explain that “parasitic capacitance of the conductive yarns is an intrinsic property and is independent of the application frequency.” If my understanding is correct, can I make an assumption that “once the parasitic capacitance has been identified using the frequency range reported in the manuscript, we can use this to develop an application operated at 868MHz, 2.45GHz and 5.8GHz?”

2.Does parasitic capacitance impact Q-factor of the coil? If so, by how much and what is significance of this impact?

3.The second question from the previous review has yet to be fully answered: “Following the Q1, how significant the parasitic capacitance is in e-textile application and what impact the parasitic capacitance will be? The paper should compare and analyse in a case study to demonstrate the importance of parasitic capacitance in e-textile sector. The significance needs to be quantified so the method can benefit a wider community.”

The authors focus on the difference between resonant frequencies with and without considering the parasitic capacitance. This is a reasonable consideration but the impact of this difference has not been answered. Why we care about this difference and whether this would cause significant impact to a RF application. Authors have included a case study of using an NFC antenna but should consider this impact

by comparing the performance of the NFC tag (NOT NFC antenna) to investigate the significance of parasitic capacitance. The results need to demonstrate whether the parasitic capacitance is of importance.

4. Most manufacturers have a frequency tolerance for their NFC chip and if the difference is within this tolerance, the NFC chip can perform with no issue. For example, input capacitance of NXP NFC chip could have a tolerance of several pF whereas the parasitic capacitance is only hundreds of fF which is below this tolerance. In this case, the parasitic capacitance will not make a huge impact to the application and therefore it would be neglected.

The parasitic capacitance would be further reduced as the increase in the application frequency due to reduced size of antenna. Based on the theory proposed in the manuscript where parasitic capacitance is directly proportional to the length of the wire (in table 3), the capacitance would be less and therefore it could also be neglected. For example, the overall length of conductive yarn for a 2.54GHz application will be shorter than that of 13.56MHz and therefore the parasitic capacitance is less noticeable.

5. It is also confusing with the wording "NFC tag." There is no NFC tag being made in the manuscript but a coil antenna for NFC chip. An NFC tag consists of an antenna and an NFC chip so the terminology needs to be clarified and amended to avoid further confusion. More information about terminology can be found from here: https://www.st.com/content/st_com/en/support/learning/essentials-and-insights/connectivity/nfc/nfc-chips.html

6. In addition, validation of NFC tag has yet to be achieved and please refer to comment 3 and 4.

7. Fig 4 d) needs to be amended. Caption says "real and imaginary part of the impedance" but it does not show in the graph. It is the total impedance so authors need to extract/measure real and imaginary parts of their coils. It has also brought up a further question: is the reactance more capacitive or inductive?

8. It is understood that the coil is made using an embroidery machine. Please note that the machine is able to produce patterns using different stitches which in turn, would change the capacitance. The paper does not mention the stitch used nor its supporting document. This is particularly important for the research community to re-create the coil and then use your proposed method.

Reviewer #3 (Remarks to the Author):

Thank you for your replies to my comments and questions!

Many of the questionmarks were resolved. I am still a bit unsure if the, now modified, equivalent circuit

in Figure 1c can explain the origin of the parasitic capacitances you measure. I was under the impression that both ends to the left in that figure were connected to the same potential (e.g. V_{source}) and both ends to the right were connected to the same potential (e.g. reference ground). Contrary to your explanation in your reply, where only the upper filament is connected to V_{source} and the lower to reference ground. What I wonder is: how long does it take for two filaments with a certain resistivity, that are separated along their axes for approximately 200 μm , when their left-hand ends are connected to one common potential and their right-hand ends to another potential to have the same potential gradient along their axes?

In your reply to me you write about it:

"Its purpose is to help the reader understand why parasitic capacitance exists on a piece of conductive yarn. On the other hand, the turn-to-turn model shown in Figure 3 and the lump-sum model in Figure 2 are models based on the helical inductor made from the conductive yarn. Therefore, we cannot use the model in Figure 1 to understand the lump-sum and turn-to-turn models."

This still makes me wonder if the parasitic capacitances that you measure might have another origin.

REPLY TO REVIEWER COMMENTS (Round 2)

Reviewer #1 (Remarks to the Author):

The authors' dedication to the revision of the proposed method has ensured that it is now fully trusted. I thank the authors for their efforts. I recommend publication of this paper.

Reply: Thank you for taking the time to provide us with your review comments and recommendations. Your feedback is greatly appreciated and is helping us improve our work.

Reviewer #2 (Remarks to the Author):

Authors have made amendments to the manuscript to address concerns raised by the reviewers. Much of extensive work have been done to provide evidence to the satisfactory level. However, some concerns have not been understood and explained properly and therefore the reviewer has listed them as follow:

1. This question has not been explained fully. “The paper claims that parasitic capacitance greatly affects the performance of e-textile device in RF and high frequency applications but fails to explore in such frequency bands. The maximum frequency explored in the paper is 100MHz which is well below common RF bands. Since the capacitance is frequency dependent, the paper should investigate the impact of parasitic capacitance in the correct RF bands.”

It is understood that the coli has been evaluated within a band ranging from 100kHz to 100MHz due to its system resonance. More specifically, the coli has been evaluated from 30MHz to 80MHz. Authors explain that “parasitic capacitance of the conductive yarns is an intrinsic property and is independent of the application frequency.” If my understanding is correct, can I make an assumption that “once the parasitic capacitance has been identified using the frequency range reported in the manuscript, we can use this to develop an application operated at 868MHz, 2.45GHz and 5.8GHz?”

Reply: Thank you for your comments and further question. Your understanding is correct that the parasitic capacitance is an intrinsic property. As we have demonstrated, this parasitic capacitance is associated with the microstructure of the conductive yarns; so it is passive and frequency-independent. Therefore the capacitance extracted from lower frequency band (100kHz to 100MHz in our paper) can be used to estimate the performance in other frequency band in RF and high frequency applications.

In fact, we also think frequency-dependency problem is a very interesting issue for conductive yarns. Although the parasitic capacitance is frequency-independent, the yarn's constituent materials may be frequency-dependent, such as the filament core inside the coating layer, and the non-conducting fibers that are bundled with the conductive fibers. These materials may have different dielectric behavior under high frequency and need further investigation in follow-up research.

In the revised manuscript, we have rephrased our statements to enhance clarity and prevent any potential misinterpretation:

Line 61-64 (revised manuscript):

“As far as the authors are aware, no previous studies have attempted to create circuit models and estimate specifically the parasitic capacitance of conductive yarns, despite the significant impacts that the parasitic capacitance may have on e-Textile devices”

Line 351-355 (revised manuscript):

“Our research is the first attempt to comprehensively address, model, and quantify the parasitic capacitances of conductive yarns, which are of great theoretical importance. We have estimated that the parasitic capacitance is in the range of femtofarad (fF); this estimation can be used to elucidate and compensate design variations in higher-frequency and RF applications.”

2. Does parasitic capacitance impact Q-factor of the coil? If so, by how much and what is significance of this impact?

Reply: Thank you for your question. In our case, the parasitic capacitance of the yarn has a negligible impact on the Q-factor due to its comparably small value in the femtofarad range as compared to other factors. According to the definition of the Q-factor in an RLC circuit, i.e.,

$Q = \frac{1}{R} \sqrt{\frac{L}{C}}$, the variation in resistance (R) and inductance (L) is much greater when the number of turns changes, as compared to the variation in capacitance (C). Therefore, R and L have a more significant impact on the Q-factor.

3. The second question from the previous review has yet to be fully answered: “Following the Q1, how significant the parasitic capacitance is in e-textile application and what impact the parasitic capacitance will be? The paper should compare and analyse in a case study to demonstrate the importance of parasitic capacitance in e-textile sector. The significance needs to be quantified so the method can benefit a wider community.”

The authors focus on the difference between resonant frequencies with and without considering the parasitic capacitance. This is a reasonable consideration but the impact of this difference has not been answered. Why we care about this difference and whether this would cause significant impact to a RF application. Authors have included a case study of using an NFC antenna but should consider this impact by comparing the performance of the NFC tag (NOT NFC antenna) to investigate the significance of parasitic capacitance. The results need to demonstrate whether the parasitic capacitance is of importance.

Reply: Thank you for your feedback and suggestions for further exploration of the influence of parasitic capacitance. Our case study has demonstrated that the yarn's parasitic capacitance affects the performance of inductive-coupling between coils. Inductive coupling forms the basis of wireless power transfer, wireless-charging and NFC application.

More specifically, NFC has a relatively narrow frequency band, which centers at 13.56MHz with +/- 848KHz auxiliary carrier band, therefore, frequency shift of a few hundreds of KHz will have significant impact on the performance of inductive-coupling between coils, especially while many NFC readers are tuned at 13.56MHz from the manufactures.

As shown in the case study, our estimation on parasitic capacitance will compensate the resonant frequency shift of coils made from conductive yarns; therefore it will significantly improve the performance of e-Textile-based NFC applications.

Line 317-320 (revised manuscript):

“NFC operation relies on the inductive-coupling between the reader coil and the tag coil, shifting of resonant frequency may have significant impact on the performance of an NFC system, especially its sensitivity and reading range.”

4. Most manufacturers have a frequency tolerance for their NFC chip and if the difference is within this tolerance, the NFC chip can perform with no issue. For example, input capacitance of NXP NFC chip could have a tolerance of several pF whereas the parasitic capacitance is only hundreds of fF which is below this tolerance. In this case, the parasitic capacitance will not make a huge impact to the application and therefore it would be neglected.

The parasitic capacitance would be further reduced as the increase in the application frequency due to reduced size of antenna. Based on the theory proposed in the manuscript where parasitic capacitance is directly proportional to the length of the wire (in table 3), the capacitance would be less and therefore it could also be neglected. For example, the overall length of conductive yarn for a 2.54GHz application will be shorter than that of 13.56MHz and therefore the parasitic capacitance is less noticeable.

Reply: Thanks for your valuable feedback. NFC has a relatively narrow frequency band, which centers at 13.56MHz with +/- 848KHz auxiliary carrier band. In many cases, the NFC readers are tuned at 13.56MHz, if NFC tags' resonant frequency deviates from this frequency, the performance of NFC system, in terms sensitivity and reading range, may be compromised. Although NXP NFC chips could have a tolerance of several pF whereas the parasitic capacitance is only hundreds of fF, a close-to-perfect match of resonant frequency between NFC readers and tags is still desired in some NFC applications.

We would like to reiterate that the primary focus of our paper is proposing a method to measure the parasitic capacitance of conductive yarns. The case study we presented serves to validate the legitimacy of our findings.

It is indeed true that for higher-frequency applications, the absolute value of parasitic capacitances is reduced as the dimension of the device form-factor scales with the increase of operation frequency. However, it is also true that smaller capacitance has more significant impacts in high frequency systems as compared to low frequency systems.

5. It is also confusing with the wording "NFC tag." There is no NFC tag being made in the manuscript but a coil antenna for NFC chip. An NFC tag consists of an antenna and an NFC chip so the terminology needs to be clarified and amended to avoid further confusion. More information about terminology can be found from here: https://www.st.com/content/st_com/en/support/learning/essentials-and-insights/connectivity/nfc/nfc-chips.html

Reply: We appreciate your feedback regarding the terminology used in our manuscript. We acknowledge that an NFC tag typically consists of both an antenna and an NFC chip, and we apologize for any confusion that our use of the term "NFC tag" may have caused. In our study, we replaced the NFC chip with a capacitor to simplify the experiment while maintaining the inductive-coupling nature of an NFC test system. While we believe that our approach is electrically equivalent and correct for the purposes of our research, we recognize the importance of precise terminology and will take steps to clarify this point in the manuscript to avoid any future confusion.

Line 326-327 (revised manuscript):

"To emulate the inductive-coupling property of an NFC device, we created a simulated NFC tag by connecting a fixed capacitor, electrically representing an NFC chip, to the embroidered antenna coils."

6. In addition, validation of NFC tag has yet to be achieved and please refer to comment 3 and 4.

Reply: The purpose of our case study is to demonstrate the shift of resonant frequency if NFC coil antennas are made from conductive fibers instead of traditional metallic wires. In order to test coils of different turns and structures, we replaced the NFC tag chip with a fixed capacitance, and measured only the resonant frequency of the coils. The NFC functionality was not validated. The measurement of frequency shift should suffice the purpose of parasitic capacitance estimation in this case.

Line 330-331 (revised manuscript):

"We then measured the actual resonant frequency of the tag using an NFC testing platform (QBG5C, AI YD)."

7. Fig 4 d) needs to be amended. Caption says "real and imaginary part of the impedance" but it does not show in the graph. It is the total impedance so authors need to extract/measure real and imaginary parts of their coils. It has also brought up a further question: is the reactance more capacitive or inductive?

Reply: We apologize for the confusion in Fig 4d). To clarify, the solid line represents the real part while the imaginary part is represented by the dashed line. We will update the caption accordingly. Regarding the reactance, its capacitive or inductive behavior depends on the

frequency of operation. Specifically, when the imaginary part is greater than 0, it is inductive, and when it is less than 0, it is capacitive.

Line 247-248 (revised manuscript):

“(d) Real (solid line) and imaginary (dashed line) part of the impedance of the copper wire (0.32 mm) based DUTs with different number of turns.”

8. It is understood that the coil is made using an embroidery machine. Please note that the machine is able to produce patterns using different stitches which in turn, would change the capacitance. The paper does not mention the stitch used nor its supporting document. This is particularly important for the research community to re-create the coil and then use your proposed method.

Reply: Thank you for bringing this to our attention. You are correct in that the use of different stitches would result in a change in capacitance. In our study, however, we utilized a consistent stitch size of 2 mm to fabricate the coil across all settings. We will specify this information in the Method Section.

Line 458-459 (revised manuscript):

“A consistent stitch size of 2mm is used to fabricate the coil across all settings.”

Reviewer #3 (Remarks to the Author):

Thank you for your replies to my comments and questions!

Many of the questionmarks were resolved. I am still a bit unsure if the, now modified, equivalent circuit in Figure 1c can explain the origin of the parasitic capacitances you measure. I was under the impression that both ends to the left in that figure were connected to the same potential (e.g. V_{source}) and both ends to the right were connected to the same potential (e.g. reference ground). Contrary to your explanation in your reply, where only the upper filament is connected to V_{source} and the lower to reference ground. What I wonder is: how long does it take for two filaments with a certain resistivity, that are separated along their axes for approximately 200 μm , when their left-hand ends are connected to one common potential and their right-hand ends to another potential to have the same potential gradient along their axes?

Reply: Thank you for your valuable feedback on our paper. We apologize for any confusion caused by the figure in our previous reply. As you correctly noted, both the upper and lower filaments are connected to V_{source} at the beginning (to the left) and connected to the ground at the end (to the right). In fact, Figure 1(c) represents a section of the yarn. We have included a new illustration Figure A shown below. Figure A includes 2 filaments in a conductive yarn. Initially, they have the same potential at the source. along the length of the filaments, there will be several gaps, causing the difference in length before reaching the next contact. This difference in length would make R_1 and R_4 to have different values, and the potential at two ends of C_{gap1} will be different. Therefore, there forms a parasitic capacitance between the gaps of filaments in conductive yarn. We believe that the potential gradient should be instantly established given the short length of filament. (In order to accurately calculate the time, it is necessary to determine the group velocity of the signal as it travels through the waveguide (filament). It should be noted that the group velocity of a signal in a dispersion material is dependent on its frequency. In our case, since the length is extremely small, the potential gradient can be considered to establish instantaneously.)

Figure A

Figure 1(c)

In your reply to me you write about it: "Its purpose is to help the reader understand why parasitic capacitance exists on a piece of conductive yarn. On the other hand, the turn-to-turn model shown in Figure 3 and the lump-sum model in Figure 2 are models based on the helical inductor made from the conductive yarn. Therefore, we cannot use the model in Figure 1 to understand the lump-sum and turn-to-turn models." This still makes me wonder if the parasitic capacitances that you measure might have another origin.

Reply: Thank you for your question. We appreciate the opportunity to provide further clarification. In our paper, we utilized a helical inductor to extract the parasitic capacitance of the conductive yarn. The capacitance of the helical inductor consists of two origins, namely, the structural capacitance and the parasitic capacitance. As shown in Figure 2(c), the structural capacitance consists of turn-to-turn capacitance (C_{ti}) and turn-to-ground capacitance (C_{gi}), they are not considered parasitic in our model.

The yarn's parasitic capacitance illustrated in Fig. 1 is only for one segment of the filaments, for the helical inductor, the parasitic capacitance is modeled as the capacitance C_{pgi} (as depicted in Figure 3(d)). In our paper, we believe that the parasitic capacitance is induced by the microstructure of the conductive yarn, however, if there are any other possible parasitic capacitances that may arise from the yarn, they will also be included in the C_{pgi} term in our turn-to-turn model, as shown in Figure 3(d).

Line 195-200 (revised manuscript):

“In our experiments, the number of turns of the helical inductor ranges from 8 to 20, therefore, we can ignore C_{pti} from the turn-to-turn model and only use C_{pgi} to indicate the extra parasitic capacitance induced from the microstructure of the conductive yarns. In addition, although we believe the parasitic capacitance C_{pgi} is originated from the gaps between the filaments in the conductive yarn, any other possible parasitic capacitances that may arise can also be included in C_{pgi} .”

REVIEWERS' COMMENTS

Reviewer #2 (Remarks to the Author):

Authors have made further amendments to address additional comments raised in the last round of review and I am satisfied with some answers provided. However, I am still not clear on the importance of such as a study and how this could benefit e-textile community.

Linking back to the third question I raised early regarding the quantification of NFC case study, it has demonstrated that considering additional parasitic capacitance would achieve a frequency closer to 13.56MHz. However, the benefit/improvement has yet to be fully quantified even authors responded "will significantly improve the performance of e-textile based NFC application". Some uncertainties have been shown in the response document, such as "...may have significant impact on the performance", "...may be compromised", "...a close to perfect match of resonant frequency between NFC readers and tags is still desired in some NFC applications". If the importance was quantified, all these uncertainties would have been eased. Please link to bullet point 4 of previous comment in regard to NFC chip tolerance.

In addition, the evidences shown in the supplementary document only demonstrate that the frequency variation is reduced after considering the parasitic capacitance but whether this reduction would help NFC communication is still missing. Typically, for a given NFC antenna, a circuit is used to provide impedance matching to maximising the power transfer. Therefore, I am not sure how critical the parasitic capacitance is in this scenario. I understand that NFC is just to demonstrate the proposed parasitic capacitance concept but whether this is a good example or major concern is questionable. I don't have other concerns other than the lack of demonstratable importance of proposed concept in e-textile community.

Reviewer #3 (Remarks to the Author):

Thank you for your replies to my questions, they have been answered satisfactory.

As I said in my first review, the topic is really interesting and the whole research community of smart textiles will hopefully benefit from your results.

I recommend publication of the paper.

REPLY TO REVIEWER COMMENTS (Round 3)

We appreciate the efforts of the editors and reviewers in helping us improve the quality of our paper. We are pleased to hear that our research on smart textiles has sparked interest and we hope it will contribute to the development of this field. We have included additional discussions on our results based on the feedback from the reviewers in this revised version. We hope that this version meets the publication requirements and standards.

Reviewer #2:

Authors have made further amendments to address additional comments raised in the last round of review and I am satisfied with some answers provided. However, I am still not clear on the importance of such as a study and how this could benefit e-textile community.

Linking back to the third question I raised early regarding the quantification of NFC case study, it has demonstrated that considering additional parasitic capacitance would achieve a frequency closer to 13.56MHz. However, the benefit/improvement has yet to be fully quantified even authors responded “will Significantly improve the performance of e-textile based NFC application”. Some uncertainties have been shown in the response document, such as “...may have significant impact on the performance”, “...may be compromised”, ...”a close to perfect match of resonant frequency between NFC readers and tags is still desired in some NFC applications”. If the importance was quantified, all these uncertainties would have been eased. Please link to bullet point 4 of previous comment in regard to NFC chip tolerance.

In addition, the evidences shown in the supplementary document only demonstrate that the frequency variation is reduced after considering the parasitic capacitance but whether this reduction would help NFC communication is still missing. Typically, for a given NFC antenna, a circuit is used to provide impedance matching to maximising the power transfer. Therefore, I am not sure how critical the parasitic capacitance is in this scenario. I understand that NFC is just to demonstrate the proposed parasitic capacitance concept but whether this is a good example or major concern is questionable.

I don't have other concerns other than the lack of demonstratable importance of proposed concept in e-textile community.

Reply: We appreciate your feedback and recognize your concerns regarding the potential benefit of parasitic estimation for conductive yarns, especially for the case study of embroidered NFC coils presented in our revised manuscripts. In this case study, the experiments had demonstrated that through the incorporation of parasitic capacitance, the resonant frequency can be tuned closer to the desired frequency (13.56MHz) by a few hundreds KHz.

In order to achieve higher power transfer efficiency, many NFC systems utilize coils with high Q (quality factor). One example can be seen from the work of Shahmohammadi, et. al., where the authors demonstrated a high-Q NFC system as shown in the figure below. It can be

estimated that a shift of 100 KHz in resonant frequency would result in around 20 dB loss in the reception power.

(Figure: Extracted from M. Shahmohammadi, M. Chabalko and A. P. Sample, "High-Q, over-coupled tuning for near-field RFID systems," 2016 IEEE International Conference on RFID (RFID), Orlando, FL, USA, 2016, pp. 1-8, doi: 10.1109/RFID.2016.7488016.)

Two factors determine the extent of impact of parasitic capacitance on the performance of e-Textile devices in high-frequency applications, i.e., the operating frequency and the Q-value. On one hand, high frequency requires smaller capacitance in LC resonant circuits, where a small amount of capacitance variation will cause significant frequency shift. On the other hand, higher Q value creates narrow band-width; a small frequency shift from the peak (S11, for example) will cause significant power loss, as demonstrated in the work of Shahmohammadi, et. al.

(Figure: Extracted from M. Shahmohammadi, M. Chabalko and A. P. Sample, "High-Q, over-coupled tuning for near-field RFID systems," 2016 IEEE International Conference on RFID (RFID), Orlando, FL, USA, 2016, pp. 1-8, doi: 10.1109/RFID.2016.7488016.)

Line 288 to 293 (revised manuscript):

“Two factors determine the extent of impact of parasitic capacitance on the performance of e-Textile devices in high-frequency applications, i.e., the operating frequency and the Q-value. On one hand, high frequency requires smaller capacitance in LC resonant circuits, where a small amount of capacitance variation will cause significant frequency shift. On the other hand, higher Q value creates narrow band-width; a small frequency shift from the peak (S_{11} , for example) will cause significant power loss .”

Reviewer #3:

Thank you for your replies to my questions, they have been answered satisfactory. As I said in my first review, the topic is really interesting and the whole research community of smart textiles will hopefully benefit from your results. I recommend publication of the paper.

Reply: Thank you for your positive feedback and recommendation to publish our paper. We are glad to hear that our research on smart textiles is of interest and we hope it will contribute to the advancement of this field. We appreciate your support and thank you for taking the time to review our work.